# LocoVR: Multiuser Indoor Locomotion Dataset in Virtual Reality

**Kojiro Takeyama**[1,2], **Yimeng Liu**[1], **Misha Sra**[1]
1: University of California Santa Barbara, 2: Toyota Motor North America
`{takeyama,yimengliu,sra}@ucsb.edu`

## Abstract

Understanding human locomotion is crucial for AI agents such as robots, particularly in complex indoor home environments. Modeling human trajectories in these spaces requires insight into how individuals maneuver around physical obstacles and manage social navigation dynamics. These dynamics include subtle behaviors influenced by proxemics - the social use of space, such as stepping aside to allow others to pass or choosing longer routes to avoid collisions. Previous research has developed datasets of human motion in indoor scenes, but these are often limited in scale and lack the nuanced social navigation dynamics common in home environments. To address this, we present LocoVR, a dataset of 7000+ two-person trajectories captured in virtual reality from over 130 different indoor home environments. LocoVR provides accurate trajectory data and precise spatial information, along with rich examples of socially-motivated movement behaviors. For example, the dataset captures instances of individuals navigating around each other in narrow spaces, adjusting paths to respect personal boundaries in living areas, and coordinating movements in high-traffic zones like entryways and kitchens. Our evaluation shows that LocoVR significantly enhances model performance in three practical indoor tasks utilizing human trajectories, and demonstrates predicting socially-aware navigation patterns in home environments. The dataset and evaluation code are available at https://github.com/kt2024-hal/LocoVR.

## 1 Introduction

Predicting human trajectories is crucial for AI systems like home robots. While many outdoor pedestrian trajectory datasets exist, they are not applicable to indoor settings due to differences in geometric complexity, scale, and movement patterns. An ideal indoor dataset would include diverse scenes and trajectories, but creating such a dataset at scale is challenging. Camera-based collection methods often fail due to obstructions, while advanced 3D scanning methods are limited by high costs and time constraints. Consequently, a comprehensive dataset of human locomotion in varied indoor environments remains elusive, hindering the development of AI systems that can effectively navigate and assist in home settings.

To overcome data collection challenges, we propose LocoVR, a dataset captured in virtual reality (VR) that efficiently captures detailed spatial information, human-scene interactions, and human-human social motion behaviors across diverse indoor environments. LocoVR captures task-focused movements of two people in over 130 home settings, including their trajectories, head orientations, and precise spatial data. Crucially, LocoVR captures motion proxemics - the social use of space, such as yielding in narrow spaces, maintaining personal distances in shared areas, and coordinating movements in high-traffic considering the Interpersonal Adaptation Theory Burgoon et al. (1995). These proxemics-based motion behaviors, often missing in current datasets, serve as a form of non-verbal communication, and are influenced by factors such as the relationship between individuals and their cultural backgrounds Hall (1963); Watson (2014). Human social dynamics can provide valuable insights for home robots to navigate domestic spaces more naturally while adhering to implicit social norms.

Our goal is to understand and predict human trajectories in complex indoor environments by considering both geometric constraints and social proxemics. Geometrically, we aim to model how people

avoid obstacles and find efficient paths. Socially, we want to capture how individuals anticipate and react to other people's movements, adjusting their trajectory to avoid collisions, maintaining personal space, and minimizing path interference.

We demonstrate our dataset through three tasks: global path prediction, trajectory prediction, and goal area prediction (Figure1). The first two tasks showcase the dataset's capability to facilitate geometrically and socially aware path predictions, while the last task demonstrates its versatility in supporting a broad spectrum of applications. Our key contributions are outlined as follows.

1. Developing a VR system for the efficient and accurate collection of two-person trajectories across diverse indoor environments.

2. Building the first large-scale indoor trajectory dataset featuring two-person motions, which enhances task performance in unseen indoor scenes from both geometric and social perspectives.

3. Showcasing enhanced model performance trained on our dataset across three practical indoor tasks, demonstrating geometrically and socially aware navigation patterns in home environments.

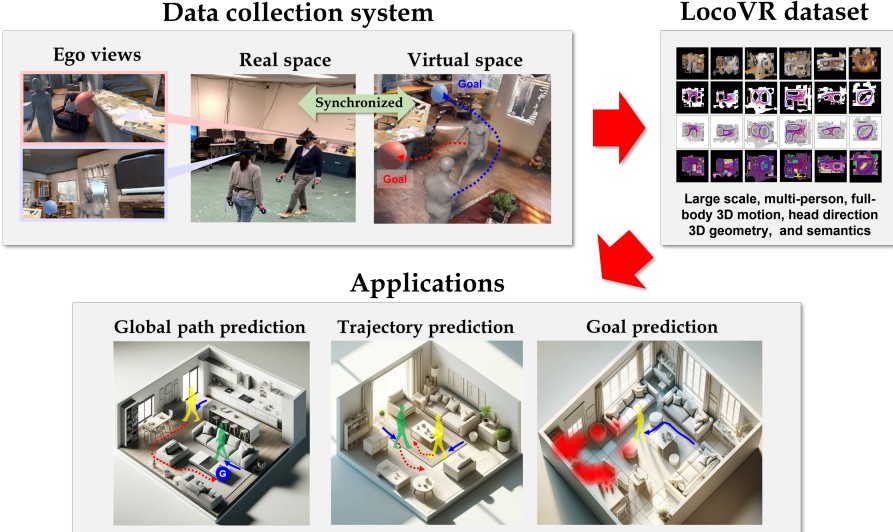

Figure 1: Overview of the data collection VR system, snippets of the LocoVR dataset, and the dataset applications. We collected multi-person trajectories in 131 complex indoor scenes, including trajectory with head orientation, precise geometry. In the dataset applications, predicted and past trajectories are shown in red and blue, respectively.

## 2 RELATED WORK

### 2.1 DATASETS FOR SOCIAL NAVIGATION

Datasets for social robot navigation can be categorized based on their scene types and the nature of the robot or human agent involved. Studies such as Rudenko et al. (2020); Schreiter et al. (2024); Finean et al. (2023); Manso et al. (2020) provide datasets collected in controlled lab settings where robots coexist with humans. While these datasets offer valuable insights into human-robot interactions in indoor settings, their scene structure is simplistic, and the number of unique scenes is limited, which reduces their capability on diverse real-world indoor settings. On the other hand, datasets like SIT Bae et al. (2024), SACSON Hirose et al. (2023), JRDB Vendrow et al. (2023), and Socnav1 Manso et al. (2020) capture interactions in crowded public spaces by robots or human equipped with cameras and Lidar. Despite their richness in capturing diverse interactions, the open and unstructured nature of these scenes poses challenges for generalizing the findings to indoor settings. Furthermore, the partial observation of scene structures limits their application for indoor navigation tasks, such as global path planning. Datasets such as those proposed by Zhou et al. Zhou et al. (2012), Robicquet et al.

Robicquet et al. (2016), Ess et al. Ess et al. (2007), and Kothari et al. Kothari et al. (2021) used aerial cameras to capture human trajectories and surrounding scene images. They provide complementary data on human trajectories and scene geometry across a wide range of environments, enabling the study of crowd dynamics. However, their applicability to indoor scenes is limited due to differences in scene structure, spatial constraints, and interaction dynamics within enclosed spaces.

## 2.2 DATASETS FOR HUMAN MOTION SYNTHESIS AND GENERATION

Understanding human motion is important for various research problems, such as synthesizing motion for 3D environments. Prior work has studied the 3D human motion problem and built datasets to support research in this domain. One line of research has explored human-scene interactions and focused on understanding how environmental constraints affect human behavior, such as GIMO (Zheng et al., 2022; Guzov et al., 2021; Zhang et al., 2022). A challenge faced by human-scene interaction datasets is capturing scene variation is difficult and expensive. To address this challenge, prior research Wang et al. (2022); Cao et al. (2020) has worked on synthesizing human motions in virtual environments and game engines or generating human motions with generative AI models. However, these synthesized motions do not necessarily follow human behavior principles, especially when social motion behaviors are involved. Additionally, the existing datasets focus mainly on single-person motions though understanding multi-person social motion behavior is critical for many practical applications, such as human-robot collaboration. Moreover, existing human motion datasets provide limited locomotion data needed to understand room-scale human motion dynamics, as they primarily focus on capturing fine-grained details of individual actions, such as opening a fridge. To fill these research gaps, we introduce LocoVR, a novel dataset comprising two-person trajectories with social motion behaviors, collected using VR to allow a large number of trajectories across diverse indoor scene variations.

## 2.3 VR SYSTEMS FOR HUMAN MOTION ANALYSIS

VR technologies including motion tracking systems have been extensively used to study human-scene interactions and behavior across various domainsLee et al. (2023); Takahashi et al. (2021); Simeone et al. (2017); Brookes et al. (2020), including sports strategy evaluation Wang et al. (2024b), road crossing studies Gallo et al. (2024), and navigation in crowded environments Yun et al. (2024). These applications have demonstrated their versatility in replicating real-world scenarios while ensuring safety and repeatability. Datasets specifically developed for VR-based studies focus on distinct scenarios, such as road crossing Wu et al. (2023) and object-approaching tasks Araújo et al. (2023). While these datasets facilitate task-specific behavioral analyses, they lack representation of complex indoor locomotion scenarios involving intricate geometry and multi-person interactions. To address this gap, we introduce a VR system that delivers an immersive locomotion experience across diverse indoor scenes, offering a novel dataset and valuable insights to advance research on human locomotion and interactions in complex indoor environments.

# 3 LOCOVR DATASET

## 3.1 OVERVIEW

Table 1 summarizes the statistics of existing human trajectory and motion datasets and LocoVR. Our dataset contains 2500K frames of human trajectories in 131 scenes (see Figure 2 for examples). The number of scenes surpasses all the real human motion datasets. We collected two-person trajectories that are geometrically and socially aware, which is not included in most of the compared datasets. The number of trajectories is 7071 in total (see Appendix I for detailed statistics of LocoVR). LocoVR facilitates the enhancement of task performances from both geometric and social perspectives in unseen, complex, and confined indoor environments. Also, it includes body tracker data on head/waist/hands/feet as auxiliary information. These additional observations could facilitate a deeper understanding of human locomotion and enhance model performance.

## 3.2 DATA COLLECTION

Figure 1 shows our data collection system. The data collection experiment has been approved by the IRB at our institution under protocol number 14-22-0646. During the experiment, two people

Table 1: Statistics of existing human motion datasets and our LocoVR dataset.

| Dataset | Frame | Scene | | | Subject | | | Target − action |
| | | Count | Geometry | Location | Pos/Pose | Multi | Motion* | |
|---|---|---|---|---|---|---|---|---|
| HPS (Guzov et al., 2021) | 300K | 8 | ✓(3D mesh) | Out/Indoor | 3D | ✓ | Real | Daily actions |
| EgoBody (Zhang et al., 2022) | 153K | 15 | ✓(3D mesh) | Indoor | 3D | ✓ | Real | Daily actions |
| PROX (Hassan et al., 2019) | 100K | 12 | ✓(3D mesh) | Indoor | 3D | | Real | Daily actions |
| GIMO (Zheng et al., 2022) | 129K | 19 | ✓(3D mesh) | Indoor | 3D, Gaze | | Real | Daily actions |
| Grand Station (Zhou et al., 2012) | 50K | 1 | ✓(Aerial image) | Outdoor | 2D | ✓ | Real | Trajectory |
| SDD (Robicquet et al., 2016) | 929K | 6 | ✓(Aerial image) | Outdoor | 2D | ✓ | Real | Trajectory |
| ETH (Ess et al., 2007) | 50K | 2 | ✓(Aerial image) | Outdoor | 2D | ✓ | Real | Trajectory |
| THOR (Rudenko et al., 2020) | 360K | 3 | ✓(3D point cloud) | Indoor | 2D | ✓ | Real | Trajectory |
| JRDB (Vendrow et al., 2023) | 636K | 30 | ✓(3D point cloud) | Out/Indoor | 3D | ✓ | Real | Trajectory |
| GTA-IM (Cao et al., 2020) | 1000K | 10 | ✓(3D mesh) | Indoor | 3D | | Synthetic | Trajectory |
| HUMANISE (Wang et al., 2022) | 1200K | 643 | ✓(3D mesh) | Indoor | 3D | | Synthetic | Daily actions |
| CIRCLE (Araújo et al., 2023) | 4300K | 9 | ✓(3D mesh) | Indoor | 3D | | Real | Daily actions |
| THOR-MAGNI (Schreiter et al., 2024) | 1260K | 4 | ✓(3D mesh) | Indoor | 3D, Gaze | ✓ | Real | Trajectory, Daily actions |
| LocoVR (Ours) | 2500K | 131 | ✓(3D mesh) | Indoor | 3D, Head | ✓ | Real | Trajectory, Social motion |

*"Real" refers to real human motions and walking behaviors captured via video or motion capture; "Synthethic" refers to synthesized human motions and behaviors via animation techniques.

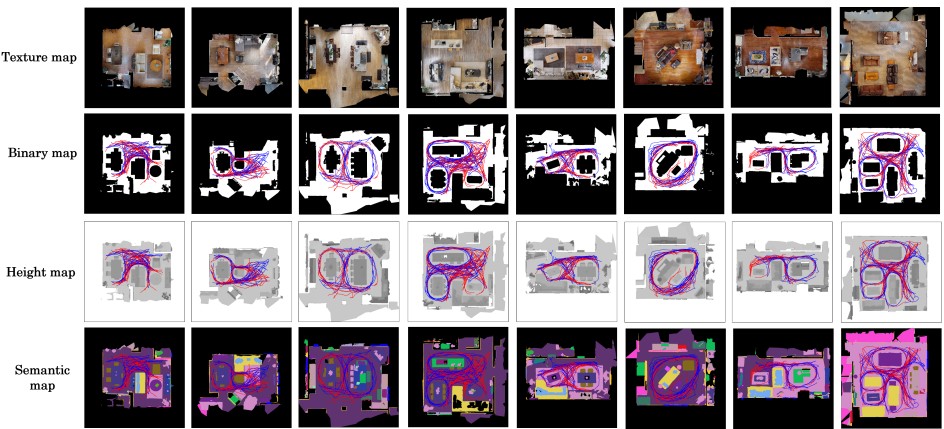

Figure 2: LocoVR includes 131 scenes with detailed spatial information, like photorealistic textures, 3D geometry, and semantics. Blue and red curves show two people's trajectories in one session.

wore VR headsets to see a shared virtual environment where they could interact with each other and perform tasks that required walking. The participants' movements were recorded in real-time by motion capture and mapped onto virtual avatars that move in the same way as the participants to help them keep social awareness. The advantages of our system include: (1) scene variation is fast and easy by switching the virtual scenes such that human trajectory data in a large variety of scenes can be collected; (2) accurate spatial information can be obtained by recording the spatial data in digital format; (3) participants can walk naturally and produce locomotion data recorded in VR as the virtual room has been well aligned with the physical space; (4) geometrically and socially aware motion behaviors can be accurately and easily controlled and replicated in a virtual space.

The data collection was conducted in a 10m by 10m open indoor room in the physical space, where participants walked within similarly sized virtual rooms. In the experiment, each person was assigned a unique goal, represented by a virtual marker that only they could see. Once they reached their goal, a new goal appeared in a different location, starting a new round of the task. The new goal appeared at a random location, which was at least 2m away from the previous goal position, to prevent short trajectories. This process repeated around 5 minutes per scene to allow the participants to fully explore the virtual environment and add variations to the collected trajectory data. See Appendix F for more details of the data collection setup.

## 4 EVALUATION

We evaluate our dataset by demonstrating its usage in the following trajectory-based indoor tasks:

1. **Global path prediction:** This task estimates a static global path from a start to a goal, applicable to goal-conditioned human path prediction or global path planning for robots. The task demonstrates the capability of our dataset to learn physically and socially plausible human trajectory, including social motion behaviors such as maintaining social distance when passing or choosing longer routes to avoid others. The input includes the past trajectories of two people, $p_1$ and $p_2$ (length=1.0s, interval=0.067s), past heading directions of $p_1$ and $p_2$ (length=1.0s, interval=0.067s), scene map, and goal position. The output is a static path from the start to the goal.

2. **Trajectory prediction:** This task predicts short-term future trajectories based on past movement data and is used to develop policies for robots to avoid collisions with other humans. By training on our dataset, trajectory prediction models can consider the movements of others in indoor environments where the space is small and where obstacles have a significant impact on path choice. The input contains the past trajectories of $p_1$ and $p_2$ (length=1.0s, interval=0.067s), past heading directions of $p_1$ and $p_2$ (length=1.0s, interval=0.067s), and the scene map. The output is time-series future trajectories of $p_1$ and $p_2$ (length=3.0s, interval=0.067s).

3. **Goal prediction:** This task predicts the goal position based on past trajectory and can be used in applications where robots or AI predict potential goals as people begin to move toward their next task location. We demonstrate that the goal prediction model trained on our dataset effectively narrows down the goal candidates by considering scene geometry and past trajectory. The input are the past trajectory and heading directions of $p_1$ (length=6.0s, interval=0.067s) and the scene map. The output is a predicted goal position. Note that the arrival time at the goal is not given.

In these tasks, it is crucial to consider how human trajectories are influenced by goal positions, the movements of other people, and scene geometry. Particularly, geometry is a dominant factor affecting human trajectories in complex indoor environments. Although there are geometry-aware trajectory prediction models like NSP Yue et al. (2022), Goal-GAN Dendorfer et al. (2020), and SoPhie Sadeghian et al. (2019), they often compress geometric features into small sets, losing the detailed structure of the entire scene. As a result, these models struggle to learn how humans move in complex indoor geometries. To address this, we employed U-Net-based models (a simple U-Net and Ynet(Mangalam et al., 2021)) to preserve the scene geometry. Details of the benchmark models for each task are described in the following sections. Additionally, in the tasks, we incorporated both the trajectories and heading directions (front direction of head poses) of individuals to maximize prediction performance, as head direction data are available in all benchmark datasets, including LocoVR, GIMO, and THOR-MAGNI.

### 4.1 DATASETS

#### 4.1.1 TRAINING DATASETS

We evaluated LocoVR and two existing datasets as a benchmark. Given the absence of datasets specifically focusing on locomotion in complex indoor scenes, we chose GIMO and THOR-MAGNI as training data because they closely align with LocoVR and are suitable for our tasks.

**LocoVR:** LocoVR is our main contribution, and it was collected using our VR system. The dataset includes over 7000 trajectories in 131 indoor environments. We split it into training (85%) and validation sets (15%).

**GIMO Zheng et al. (2022):** GIMO is an indoor daily activity dataset containing trajectory data with heading information in real complex indoor environments, while it has limited scene variations and only single-person data. We extracted the locomotion data and excluded trajectories that were too short ($< 2s$), resulting in 187 trajectories in 19 scenes. We divided the dataset into training (85%) and validation sets (15%).

**THOR-MAGNI Schreiter et al. (2024):** THOR-MAGNI is an indoor multi-person trajectory dataset that contains a number of trajectories comparable to our dataset, including heading information. However, it includes only four types of scene maps, which are similar to each other. To align the data format with our test data, we extracted trajectory segments between goals and then picked up pairs of

trajectories from multi-person trajectories within the same scene. We excluded short trajectories ($<$ 2s) and trajectory pairs that included a time jump in either trajectory. As a result, we obtained around 10,000 trajectories in 4 scenes and divided them into training (85%) and validation sets (15%).

### 4.1.2 Testing dataset

**LocoReal:** To test the models on real-world data, we built a human trajectory dataset collected in a physical room space. To collect this dataset, we invited two participants to walk in a room that contained furniture. Each participant's movements were tracked and recorded by a motion capture system. They were then given a list of goals and asked to reach each goal in sequence, one after another. This allowed us to record the full trajectories of their motion including social behaviors necessary to navigate around the furniture and in 4 tight spaces. See Appendix G for the details.

### 4.2 Implementation details

In our evaluation, all the positional information, such as trajectory or goal position, is handled in 2D images. We use binary maps as scene maps, with 1 indicating the area between -0.3m to 0.3m height above the floor level and 0 for all the other areas. The map size is 256 pixels by 256 pixels in the image and 10m by 10m in the physical world. All the training data are augmented by horizontal flipping and rotation with 90, 180, and 270 degrees, increasing the number of training data by eight-fold. We also augmented the data in a time-series direction to obtain different sets of past trajectories and ground truth future trajectories. We define a fixed length time window and slide it with a certain interval to obtain sets of augmented trajectories. See Appendix H for the details.

### 4.3 Global path prediction

#### 4.3.1 Benchmark models

**A\* (Hart et al., 1968) + U-Net Ronneberger et al. (2015):** A\* is an algorithm that finds the optimal path using a cost map, minimizing the total cost from the start to the goal. While it is commonly used for robots to find the shortest path to the goal, we use it to find a human-like path by incorporating the cost map created based on probabilistic path distributions derived from a model trained on the datasets. Specifically, we trained the model with a simple U-Net structure using the training datasets (LocoVR, GIMO, THOR-MAGNI). The model's input and output have been explained in Section 4. To obtain the cost map, first take the reciprocal of the model's output and then multiply it by the obstacle map to incorporate obstacle information. Additionally, we used two types of non-learning-based cost maps as benchmarks: MAP and DISTMAP. MAP is based on the scene map, where the cost is 1 in obstacle areas and 0 elsewhere. DISTMAP is based on the distance from the nearest obstacle, defining the cost as $1/(1+d)$, where $d$ is the distance from the obstacle in pixels (with a maximum value of $d = 10$).

**Ynet (Mangalam et al., 2021):** Ynet is a state-of-the-art technique for goal-conditioned human trajectory prediction. It can generate long-term trajectories considering complex scene geometry based on multiple U-Net framework. While Ynet predicts a dynamic trajectory to the goal, we convert into a static global path by projecting the trajectory onto an image to match the benchmark's output.

#### 4.3.2 Metrics

We adopted an evaluation metric based on Chamfer distance to assess the differences between the predicted and ground-truth paths. This metric calculates the distance from each pixel in the predicted path to the nearest pixel in the ground-truth path. We employed the mean and maximum of these distances across all pixels in the path and converted them to meters.

#### 4.3.3 Results

**Quantitative results:** Table 2 presents the evaluation results based on two metrics. A\* with the U-Net trained with LocoVR demonstrated significantly superior performance compared to the benchmarks that include models trained on GIMO and THOR-MAGNI. It is also shown that Ynet trained on LocoVR presents better accuracy than the other two datasets. This is mainly due to the difference in scalability of the datasets, including a number of trajectories and scenes. In GIMO, both the number of trajectories and scenes are limited, whereas THOR-MAGNI has many trajectories but includes

only 4 scenes, leading to low generalization ability to new scenes not present in the training data. On the other hand, LocoVR ensures high performance even in unseen scenes owing to its high scalability in data amount and scene variation. See Table 5 in Appendix C for details.

Table 2: Mean and Max Chamfer distance between predicted and ground-truth paths grouped by distance to the goal. The table reports averaged value over three trials $\pm$ SD.

| Method | Mean | | | Max | | |
|---|---|---|---|---|---|---|
| | $0m \leq d \leq 3m$ | $3m \leq d \leq 6m$ | $6m \leq d$ | $0m \leq d \leq 3m$ | $3m \leq d \leq 6m$ | $6m \leq d$ |
| Ynet (GIMO) | **0.08**$_{\pm \textbf{0.003}}$ | 0.22$_{\pm 0.012}$ | 0.51$_{\pm 0.011}$ | **0.17**$_{\pm \textbf{0.003}}$ | 0.46$_{\pm 0.022}$ | 1.11$_{\pm 0.016}$ |
| Ynet (THOR-MAGNI) | 0.10$_{\pm 0.003}$ | 0.30$_{\pm 0.006}$ | 0.65$_{\pm 0.014}$ | 0.19$_{\pm 0.004}$ | 0.56$_{\pm 0.008}$ | 1.29$_{\pm 0.023}$ |
| Ynet (LocoVR) | 0.09$_{\pm 0.002}$ | **0.18**$_{\pm \textbf{0.004}}$ | **0.42**$_{\pm \textbf{0.050}}$ | 0.18$_{\pm 0.002}$ | **0.37**$_{\pm \textbf{0.005}}$ | **0.92**$_{\pm \textbf{0.089}}$ |
| A* + MAP | 0.10$_{\pm 0}$ | 0.27$_{\pm 0.000}$ | 0.40$_{\pm 0.000}$ | 0.22$_{\pm 0.000}$ | 0.58$_{\pm 0.000}$ | 0.89$_{\pm 0.000}$ |
| A* + DISTMAP | 0.102$_{\pm 0}$ | 0.18$_{\pm 0.000}$ | 0.26$_{\pm 0.000}$ | 0.24$_{\pm 0.000}$ | 0.46$_{\pm 0.000}$ | 0.66$_{\pm 0.000}$ |
| A* + U-Net (GIMO) | 0.09$_{\pm 0.002}$ | 0.23$_{\pm 0.006}$ | 0.36$_{\pm 0.011}$ | 0.20$_{\pm 0.004}$ | 0.53$_{\pm 0.013}$ | 0.84$_{\pm 0.024}$ |
| A* + U-Net (THOR-MAGNI) | 0.07$_{\pm 0.001}$ | 0.21$_{\pm 0.007}$ | 0.30$_{\pm 0.005}$ | 0.17$_{\pm 0.001}$ | 0.45$_{\pm 0.014}$ | 0.71$_{\pm 0.015}$ |
| A* + U-Net (LocoVR) | **0.06**$_{\pm \textbf{0.001}}$ | **0.12**$_{\pm \textbf{0.002}}$ | **0.19**$_{\pm \textbf{0.003}}$ | **0.15**$_{\pm \textbf{0.001}}$ | **0.29**$_{\pm \textbf{0.004}}$ | **0.50**$_{\pm \textbf{0.014}}$ |

**Qualitative results:** Figure 3 shows the result of global path prediction by A* with U-Net trained on each dataset, in four different scenes. In each image, the yellow distribution indicates lower values in the cost map that guides the global path prediction. The green and blue lines represent the past trajectories of $p_1$ and $p_2$, respectively. The orange circle indicates $p_1$'s goal position. The light green and red lines denote the groundtruth and predicted global path of $p_1$.

While the cost maps of learning-based methods emphasize expected future paths regions, those in GIMO and THOR-MAGNI are not clear and continuous, hindering the prediction of smooth, human-like paths. This limitation stems from the restricted scene variations in GIMO and THOR-MAGNI, leading to poor performance in unseen environments. In contrast, the LocoVR dataset, with its large-scale diversity, enables the prediction of geometry-aware smooth paths, even in complex and previously unseen environments.

Furthermore, LocoVR also demonstrates its ability to predict social motion behaviors. In scene 1 and 2 where $p_2$ is walking on the $p_1$'s shortest route to the goal, only the model with LocoVR accurately predicts a detour route to avoid interrupting $p_2$. In contrast, other models predict shorter routes that lead to potential collisions with $p_2$. In scenes 3 and 4, where $p_1$ and $p_2$ pass each other in a narrow space, the model trained on LocoVR predicts paths that maintain a social distance from $p_2$'s potential trajectory, closely matching the ground truth. Specifically, in scene 3, $p_1$ curves closer to the wall to keep distance from $p_2$, while in scene 4, $p_1$ steps aside to create space for $p_2$ to pass. This is attributed to LocoVR's capability to learn social motion behaviors across diverse scenes.

## 4.4 TRAJECTORY PREDICTION

### 4.4.1 BENCHMARKS

**U-Net Ronneberger et al. (2015):** We evaluated a simple model with a U-Net structure trained using the datasets (LocoVR, GIMO, THOR-MAGNI). Past trajectories and heading directions of $p_1$ and $p_2$ and the scene map are concatenated and fed to the model, then the U-Net structured encode-decoder outputs probabilistic distribution of dynamic trajectory for $p_1$ and $p_2$ represented by images.

**Ynet (Mangalam et al., 2021):** Here, we evaluate performance on dynamic trajectory prediction using Ynet trained on the datasets (LocoVR, GIMO, THOR-MAGNI). Ynet is a single-person trajectory predictor; we use the past trajectory of $p_1$ and the scene map as the input. The output is the probabilistic distribution of dynamic trajectory for $p_1$ represented by images.

### 4.4.2 METRICS

We use ADE (Average Displacement Error), a commonly used metric, to evaluate the performance of trajectory synthesis. ADE refers to the mean squared error over all the time correspondence points on predicted and ground-truth trajectories. The ADE scale is represented in meter units.

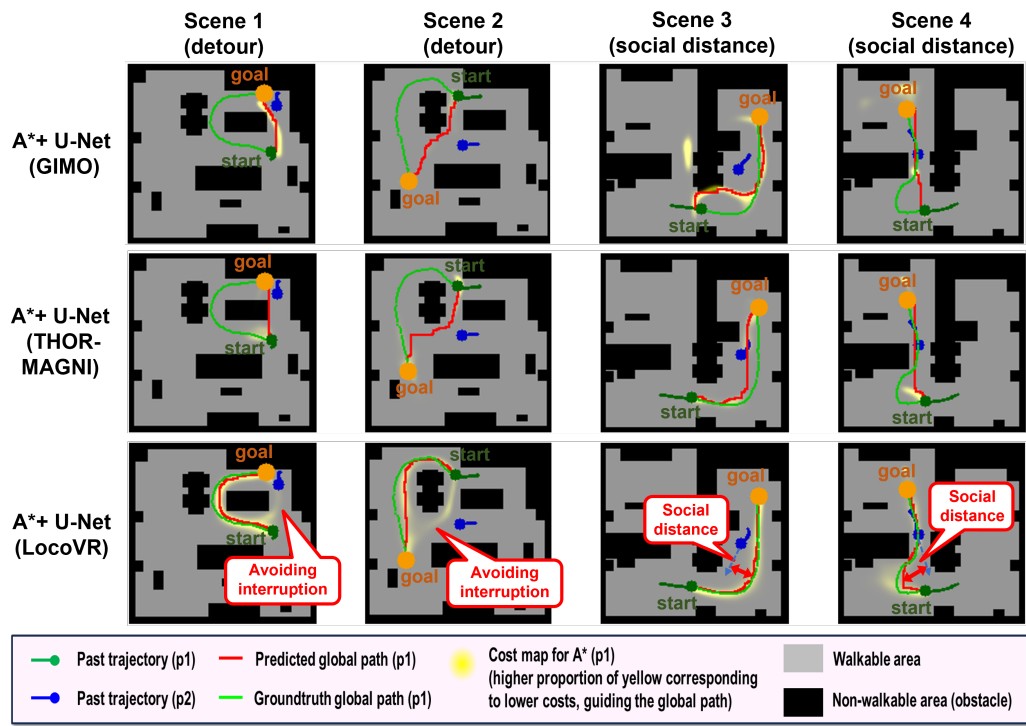

Figure 3: Predicted global paths with cost maps (intense yellow represents low-cost areas). A* generates the optimal path that minimizes the cost along the way. With LocoVR, the cost map concretely guides human-like paths (red line) that are capable of avoiding collision with obstacles and other people's paths, which align with the groundtruth paths (green line).

### 4.4.3 RESULTS

**Quantitative results:** Table 3 reports the performance of trajectory prediction over time. As can be seen, both Ynet and U-Net trained on LocoVR outperform those trained on other datasets. This is mainly due to the difference in scalability as described in the global path prediction section: fewer scene variations or the limited number of trajectories with GIMO and THOR-MAGNI result in a lack of generalization performance on new scenes. Additionally, social motion behavior, which is not contained in GIMO, is a factor that affects performance in the two-person setting. See Table 6 in Appendix C for details.

Table 3: ADE (Average Displacement Error) between predicted and ground-truth time-series trajectories. The table reports averaged error [m] in all of the trajectories over three trials $\pm$ SD.

| $Method$ | $0s \leq t \leq 1s$ | $1s \leq t \leq 2s$ | $2 \leq t \leq 3s$ |
|---|---|---|---|
| Ynet (GIMO) | $0.28_{\pm 0.010}$ | $0.53_{\pm 0.011}$ | $0.81_{\pm 0.013}$ |
| Ynet (THOR-MAGNI) | $0.62_{\pm 0.026}$ | $0.89_{\pm 0.052}$ | $0.88_{\pm 0.042}$ |
| Ynet (LocoVR) | $\mathbf{0.21}_{\pm \mathbf{0.006}}$ | $\mathbf{0.40}_{\pm \mathbf{0.013}}$ | $\mathbf{0.61}_{\pm \mathbf{0.011}}$ |
| U-Net (GIMO) | $0.19_{\pm 0.009}$ | $0.34_{\pm 0.010}$ | $0.55_{\pm 0.013}$ |
| U-Net (THOR-MAGNI) | $0.59_{\pm 0.004}$ | $0.92_{\pm 0.011}$ | $1.14_{\pm 0.016}$ |
| UNet (LocoVR) | $\mathbf{0.11}_{\pm \mathbf{0.000}}$ | $\mathbf{0.24}_{\pm \mathbf{0.004}}$ | $\mathbf{0.44}_{\pm \mathbf{0.010}}$ |

**Qualitative results:** Figure 4 shows the result of trajectory prediction with U-Net. The model trained on LocoVR is able to predict a trajectory, taking into account both the obstacles and the other person's movement. In contrast, predicted trajectory distribution with GIMO is spread to multiple directions, resulting in collisions with other people since GIMO does not include multi-person data. With THOR-MAGNI, the predicted trajectory becomes stuck along the way due to its unstable performance in unseen scenes.

Figure 4: Predicted trajectory and probabilistic distribution using U-Net. U-Net trained on LocoVR predicts $p_1$'s trajectory that smoothly proceeds with no collision with obstacles or other people.

## 4.5 GOAL PREDICTION

### 4.5.1 BENCHMARK MODELS

**U-Net Ronneberger et al. (2015):** We applied a simple U-Net model to predict goals. Inputs are past trajectories of $p_1$ and the scene map; the output is the probabilistic distribution of $p_1$'s goal position.

**RANDOM:** We evaluated two types of random sampling of goals according to the metric defined below. One method randomly determines the goal position to assess goal position accuracy, while the other randomly selects goal objects to evaluate object prediction accuracy.

**NEAREST:** This benchmark determines the goal position based on a person's current position, using two different methods according to the metrics. One samples the goal within 1.5m from the person's current position, while the other selects goal objects based on their distance from the current position.

### 4.5.2 METRICS

**Goal position error:** It is defined as the distance between the true goal position and the predicted goal position used to measure the basic performance of goal prediction.

**Object prediction accuracy:** In the testing dataset (LocoReal), the goal position is on one of the 20+ objects in the scene map, so we evaluate the rate of predicting the correct goal object. We sampled the best three objects based on confidence and evaluated the rate it includes the true goal object.

### 4.5.3 RESULTS

**Quantitative results:** Table 4 presents the performance of the models evaluated on the two metrics. Compared to RANDOM and NEAREST, the models trained on each dataset exhibit better performance. Notably, the model trained with LocoVR significantly outperforms those trained on other datasets owing to the dataset scale. See Table 7 in Appendix C for details. Performance along the distance to the goal tends to improve as the distance decreases. Note that the narrowing of the goal area cannot be attributed to the time duration to the goal, as the arrival time is not provided in this task. It is assumed that proximity to the goal correlating with longer trajectories offer additional clues for narrowing down the goal location.

Table 4: Performance on goal position prediction and goal object prediction. The table reports averaged performance in all of the trajectories over three trials ± SD.

| Method | Goal position error | | | Object prediction accuracy | | |
|---|---|---|---|---|---|---|
| | $0m \leq d \leq 3m$ | $3m \leq d \leq 6m$ | $6m \leq d$ | $0m \leq d \leq 3m$ | $3m \leq d \leq 6m$ | $6m \leq d$ |
| RANDOM | $3.70_{\pm0.02}$ | $3.75_{\pm0.02}$ | $3.76_{\pm0.03}$ | $15.5_{\pm1.0}$ | $16.1_{\pm0.5}$ | $\mathbf{15.3_{\pm1.2}}$ |
| NEAREST | $1.76_{\pm0.00}$ | $3.89_{\pm0.00}$ | $4.73_{\pm0.00}$ | $42.7_{\pm0.0}$ | $0.5_{\pm0.0}$ | $0.0_{\pm0.0}$ |
| U-Net (GIMO) | $1.58_{\pm0.32}$ | $2.47_{\pm0.06}$ | $\mathbf{3.35_{\pm0.23}}$ | $49.2_{\pm6.7}$ | $17.8_{\pm2.0}$ | $3.9_{\pm0.8}$ |
| U-Net (THOR-MAGNI) | $1.82_{\pm0.04}$ | $3.29_{\pm0.04}$ | $4.23_{\pm0.09}$ | $40.1_{\pm1.3}$ | $18.9_{\pm0.6}$ | $9.5_{\pm1.6}$ |
| U-Net (LocoVR) | $\mathbf{0.83_{\pm0.03}}$ | $\mathbf{1.89_{\pm0.02}}$ | $3.45_{\pm0.04}$ | $\mathbf{72.2_{\pm2.6}}$ | $\mathbf{40.1_{\pm2.0}}$ | $13.5_{\pm2.7}$ |

**Qualitative results:** Figure 5 shows the results of object prediction. In LocoVR, as the trajectory progresses, the probability distribution of the goal area narrows down near the true goal object. This is due to the model learning from the dataset a policy that narrows down the goal area based on the areas already passed and the current heading direction. On the other hand, GIMO and THOR-MAGNI do not include a sufficient number of trajectories or scenes to learn a policy applicable to unseen scenes, resulting in the probability distribution of the goal area not being appropriately narrowed down.

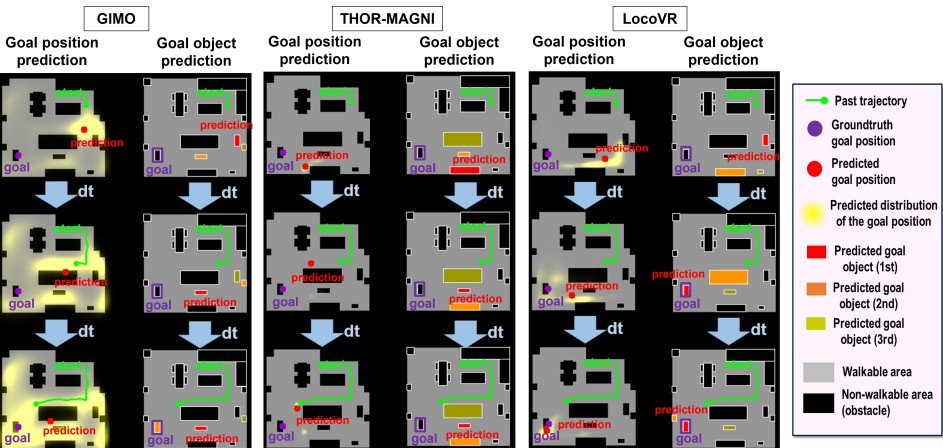

Figure 5: Predicted goal positions and objects. In the left column with LocoVR, the predicted goal area distribution (yellow color) is narrowed down as the trajectory proceeds. As a result, candidates of the goal object are accurately predicted (right column).

## 5 LIMITATIONS AND FUTURE WORK

Although we have evaluated the prediction models through the data collected in real space (LocoReal), the gaps could affect detailed human behaviors, such as walking speed or interpretation of non-verbal communication through facial expressions. Investigating these impacts and comparing LocoVR to fully synthetic datasets are important research questions for future work.

While LocoVR is designed to facilitate research on indoor human trajectories, it serves as a foundational resource for exploring the relationships between motion patterns, goal positions, and indoor scene geometries. Beyond its core purpose, LocoVR shows promise for extended tasks such as inferring indoor layouts from trajectory data or predicting human actions by incorporating trajectory Takeyama et al. (2024). Furthermore, it has potential applications in diverse fields, including robot navigation, AI agents in VR, avatar control in both real-worldSchwarz et al. (2023) and VR environmentsWang et al. (2024a), and human action predictionTakeyama et al. (2024). With its versatility, LocoVR is poised to drive innovation across these domains and beyond.

## 6 CONCLUSION

To model geometrically and socially aware human trajectories in complex indoor environments, we introduced the LocoVR dataset, which captures two-person social motion behaviors across 131 home environments, including accurate trajectory and detailed spatial information. In the experiments, we introduced three indoor tasks that utilize human trajectory: global path prediction, trajectory prediction, and goal prediction. Experimental results showed that the models trained with LocoVR outperformed other prior indoor datasets evaluated on the real-world test data. This indicates that our dataset facilitates adaptation to unseen indoor environments with complex geometries and social motion behaviors across a variety of tasks. Furthermore, these findings demonstrate the potential of virtual environments for training models that generalize well to real-world applications. We envision the data collection method to expand the variety of indoor scenes used for training and propose the experiments as a standard benchmark for future research on human motion and trajectory analysis in indoor settings.

ACKNOWLEDGMENT

This research was funded by Toyota Motor North America and conducted as a collaborative effort between its members and those from UCSB. We sincerely appreciate the members of the Human-AI Experience (HAX) Lab and other students for their participation in data collection experiments and their valuable feedback. We also extend our gratitude to the members of Toyota Motor North America, Toyota Motor Corporation, and Toyota Central Research and Development Labs Inc. for their insightful discussions and contributions. Additionally, the human subject study conducted in this research was reviewed and approved by the UCSB Human Subjects Committee under protocol number 14-22-0646.

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

## A   ETHICAL IMPLICATIONS

While our dataset provides valuable insights into adult locomotion patterns, it lacks sufficient diversity in age groups and motor abilities. This homogeneity restricts the model's generalizability to individuals outside the able-bodied adult demographic. To address this limitation, future work should focus on collecting data that encompasses a broader spectrum of ages and motor capabilities, such as children, elderly individuals, and people with mobility impairments. This will allow the model to develop a more comprehensive understanding of human movement and improve its ability to predict trajectories across a wider range of scenarios.

## B   EXPERIMENTAL DETAILS

We use the Adam optimizer (Kingma & Ba, 2014) to train the U-Net models used in the experiments. The learning rate is 5e-5, and the batch size is 16. Each model is trained for up to 100 epochs on a single NVIDIA RTX 4080 graphics card with 8G memory.

In the U-Net models, time-series trajectory is handled in a multi-channel image format. Specifically, the 2D coordinate of a position on a trajectory is plotted on a blank image (256 by 256 pixels) with a Gaussian distribution, and the time-series data is contained in multi-channels. Similarly, the goal position is encoded as an image and concatenated with the multi-channel trajectory image when fed into the model.

Further details of U-Net models are described as follows.

**Global path planning**

- Input: ($62 \times H \times W$)
  - Past trajectory of $p_1$ for 15 epochs ($15 \times H \times W$)
  - Past trajectory of $p_2$ for 15 epochs ($15 \times H \times W$)
  - Past heading directions of $p_1$ for 15 epochs ($15 \times H \times W$)
  - Past heading directions of $p_2$ for 15 epochs ($15 \times H \times W$)
  - Goal position of $p_1$ ($1 \times H \times W$)
  - Binary scene map ($1 \times H \times W$)
- Output: ($2 \times H \times W$)
  - $p_1$'s static future global path (goal conditioned) ($1 \times H \times W$)
  - $p_2$'s static future global path (non-goal conditioned) ($1 \times H \times W$)
- Groundtruth: ($2 \times H \times W$)
  - $p_1$'s static future global path ($1 \times H \times W$)
  - $p_2$'s static future global path ($1 \times H \times W$)
- Loss: BCELoss between the output and ground-truth
- U-Net channels:
  - encoder: 128, 128, 256, 256, 256
  - decoder: 256, 256, 256, 128, 128
- Calculation time for training: 10-12 hours on LocoVR

**Trajectory prediction**

- Input: ($61 \times H \times W$)
  - Past trajectory of $p_1$ for 15 epochs ($15 \times H \times W$)
  - Past trajectory of $p_2$ for 15 epochs ($15 \times H \times W$)
  - Past heading directions of $p_1$ for 15 epochs ($15 \times H \times W$)
  - Past heading directions of $p_2$ for 15 epochs ($15 \times H \times W$)
  - Binary scene map ($1 \times H \times W$)

- Output: $(90 \times H \times W)$
  - $p_1$'s future trajectory $(45 \times H \times W)$
  - $p_2$'s future trajectory $(45 \times H \times W)$
- Groundtruth: $(90 \times H \times W)$
  - $p_1$'s future trajectory $(45 \times H \times W)$
  - $p_2$'s future trajectory $(45 \times H \times W)$
- Loss: BCELoss between the output and ground-truth
- U-Net channels:
  - encoder: 128, 128, 256, 256, 256
  - decoder: 256, 256, 256, 128, 128
- Calculation time for training: 20-22 hours on LocoVR

**Goal prediction**

- Input: $(181 \times H \times W)$
  - Past trajectory of $p_1$ for 90 epochs $(90 \times H \times W)$
  - Past heading directions of $p_1$ for 90 epochs $(90 \times H \times W)$
  - Binary scene map $(1 \times H \times W)$
- Output: $(1 \times H \times W)$
  - $p_1$'s goal position $(1 \times H \times W)$
- Groundtruth: $(1 \times H \times W)$
  - $p_1$'s goal position $(1 \times H \times W)$
- Loss: BCELoss between the output and ground-truth
- U-Net channels:
  - encoder: 256, 256, 512, 512, 512
  - decoder: 512, 512, 512, 256, 256
- Calculation time for training: 30-35 hours on LocoVR

The scene map was sampled from the 3D room dataset (HM3D) by manually cutting the predefined area and projecting the height map onto the aerial-view image that covers an area of 10m by 10m. The height map is then thresholded at 0.2m and converted into the binary map representing the walkable area.

## C  ABLATION STUDY

We conducted an ablation study on LocoVR to analyze the factors that leverage the strengths of the dataset. Specifically, we imposed constraints on data scale, multi-person data usage, and heading direction usage in this study.

To investigate the impact of dataset scale, we created two types of scaled-down datasets by removing data from LocoVR, data-size-G and data-size-T. data-size-G is a dataset simulated to match the scale of GIMO, with the number of scenes and trajectories reduced to 19 and 190, respectively, by randomly selecting and removing data. data-size-T is a simulated dataset modeled after THOR-MAGNI, containing 658 trajectories across 4 scenes. This dataset has fewer trajectories than the actual THOR-MAGNI dataset because LocoVR does not have as many trajectories per scene (See fig 10 in Appendix I). However, we attempted to maximize the number of trajectories within the constraints of four scenes. In addition, we evaluated the impact on the performance by considering the other person's movement (wo/$p_2$) and heading direction (wo/head).

Table 5 presents the result of the ablation study on global path prediction. Due to the constraints, performance has deteriorated compared to the original LocoVR dataset. Notably, the reduction in dataset scale has a significant impact on performance, underscoring the importance of dataset scale in enhancing performance, which is a key strength of our dataset.

Table 6 shows the ablation study on the trajectory prediction. The performance of LocoVR deteriorated when constraints were applied to the original LocoVR, highlighting the strengths of its features.

Table 7 represents a result of the ablation study on the goal prediction. Similar to the two tasks above, the performance improvement due to LocoVR's features is also demonstrated in this table. In data-size-G, the object prediction accuracy within d< 3m is comparable to that of the original LocoVR, whereas it falls significantly short in other metrics. This is because the model trained on data-size-G tends to rely on the current location due to the lack of training data, resulting in higher accuracy when the goal is close to the current position.

Table 5: Global path prediction - Mean and Max Chamfer distance between predicted and ground-truth paths grouped by distance to the goal.

| Method | Mean | | | Max | | |
|---|---|---|---|---|---|---|
| | $0m \le d \le 3m$ | $3m \le d \le 6m$ | $6m \le d$ | $0m \le d \le 3m$ | $3m \le d \le 6m$ | $6m \le d$ |
| A* + U-Net (LocoVR data-size-G) | 0.077 | 0.194 | 0.326 | 0.180 | 0.438 | 0.752 |
| A* + U-Net (LocoVR data-size-T) | 0.076 | 0.163 | 0.242 | 0.186 | 0.417 | 0.627 |
| A* + U-Net (LocoVR wo/$p_2$) | 0.061 | 0.122 | 0.205 | 0.147 | 0.297 | 0.537 |
| A* + U-Net (LocoVR wo/head) | 0.063 | 0.134 | 0.238 | 0.150 | 0.321 | 0.614 |
| A* + U-Net (LocoVR) | **0.060** | **0.119** | **0.192** | **0.145** | **0.290** | **0.501** |

Table 6: Trajectory prediction - ADE (Average Displacement Error) between predicted and ground-truth time-series trajectories.

| Method | $0m \le d \le 3m$ | $3m \le d \le 6m$ | $6m \le d$ |
|---|---|---|---|
| U-Net (LocoVR data-size-G) | 0.274 | 0.496 | 0.775 |
| U-Net (LocoVR data-size-T) | 0.144 | 0.297 | 0.505 |
| U-Net (LocoVR wo/$p_2$) | 0.113 | 0.254 | 0.470 |
| U-Net (LocoVR wo/head) | 0.122 | 0.254 | 0.446 |
| U-Net (LocoVR) | **0.111** | **0.238** | **0.441** |

Table 7: Goal prediction - Goal position error and goal prediction accuracy.

| Method | Goal position error | | | Object prediction accuracy | | |
|---|---|---|---|---|---|---|
| | $0m \le d \le 3m$ | $3m \le d \le 6m$ | $6m \le d$ | $0m \le d \le 3m$ | $3m \le d \le 6m$ | $6m \le d$ |
| U-Net (LocoVR data-size-G) | 0.923 | 2.165 | 3.791 | **73.5** | 28.9 | 7.0 |
| U-Net (LocoVR data-size-T) | 1.552 | 2.479 | 4.118 | 63.6 | 34.8 | 6.8 |
| U-Net (LocoVR wo/head) | 1.055 | 2.151 | **3.403** | 62.9 | 25.4 | **13.6** |
| U-Net (LocoVR) | **0.83** | **1.89** | 3.45 | 72.2 | **40.1** | 13.5 |

# D  ADDITIONAL EXPERIMENTS

## D.1  TESTING ON GIMO

While the main paper utilized real-world trajectory data (LocoReal) as the test data, we conducted an additional experiment to further validate the contribution of LocoVR using an alternative test dataset. Given that THOR-MAGNI has fewer variational scenes, we selected GIMO as the test data. In this experiment, GIMO was divided into 70% for training, 15% for validation, and 15% for testing, ensuring that the scenes in the training, validation, and test sets were mutually exclusive. Additionally, to mitigate potential bias in the test data, we performed five random splits to generate five different datasets and averaged the results. Tables 8 through 10 illustrate the evaluation results for global path planning, trajectory prediction, and goal prediction, respectively. The evaluation results demonstrate that the model trained with LocoVR consistently outperforms those trained on other datasets. This superior performance is attributed to the enhanced generalization capabilities provided by LocoVR's extensive coverage of scenes and trajectories.

Table 8: Global path prediction - Mean and Max Chamfer distance between predicted and ground-truth paths grouped by distance to the goal.

| Method | Mean | | | Max | | |
|---|---|---|---|---|---|---|
| | $0m \le d \le 3m$ | $3m \le d \le 6m$ | $6m \le d$ | $0m \le d \le 3m$ | $3m \le d \le 6m$ | $6m \le d$ |
| A* + U-Net (GIMO) | $0.131_{\pm 0.0085}$ | $0.161_{\pm 0.0199}$ | $0.180_{\pm 0.0726}$ | $\mathbf{0.277}_{\pm 0.0246}$ | $0.399_{\pm 0.0314}$ | $0.517_{\pm 0.1733}$ |
| A* + U-Net (THOR-MAGNI) | $\mathbf{0.129}_{\pm 0.0106}$ | $0.151_{\pm 0.0062}$ | $0.192_{\pm 0.0742}$ | $\mathbf{0.277}_{\pm 0.0249}$ | $0.389_{\pm 0.0256}$ | $0.515_{\pm 0.1617}$ |
| A* + U-Net (LocoVR) | $\mathbf{0.129}_{\pm 0.0081}$ | $\mathbf{0.150}_{\pm 0.0113}$ | $\mathbf{0.166}_{\pm 0.0294}$ | $\mathbf{0.277}_{\pm 0.0266}$ | $\mathbf{0.384}_{\pm 0.0269}$ | $\mathbf{0.485}_{\pm 0.0851}$ |

Table 9: Trajectory prediction - ADE (Average Displacement Error) between predicted and ground-truth time-series trajectories.

| Method | $0m \le d \le 3m$ | $3m \le d \le 6m$ | $6m \le d$ |
|---|---|---|---|
| U-Net (GIMO) | $0.211_{\pm 0.0420}$ | $0.382_{\pm 0.0897}$ | $0.673_{\pm 0.1737}$ |
| U-Net (THOR-MAGNI) | $0.795_{\pm 0.0962}$ | $1.565_{\pm 0.1111}$ | $2.356_{\pm 0.1800}$ |
| U-Net (LocoVR) | $\mathbf{0.140}_{\pm 0.0209}$ | $\mathbf{0.253}_{\pm 0.0620}$ | $\mathbf{0.356}_{\pm 0.1605}$ |

Table 10: Goal prediction - Goal position error.

| Method | Goal position error | | |
|---|---|---|---|
| | $0m \le d \le 3m$ | $3m \le d \le 6m$ | $6m \le d$ |
| U-Net (GIMO) | $\mathbf{0.968}_{\pm 0.1583}$ | $2.403_{\pm 0.5629}$ | $4.446_{\pm 0.4472}$ |
| U-Net (THOR-MAGNI) | $1.766_{\pm 0.0209}$ | $3.065_{\pm 0.2679}$ | $5.332_{\pm 0.5959}$ |
| U-Net (LocoVR) | $1.054_{\pm 0.2392}$ | $\mathbf{2.100}_{\pm 0.7023}$ | $\mathbf{3.206}_{\pm 0.4658}$ |

## D.2 INFLUENCE OF SCENE INFORMATION - BINARY OBSTACLE - HEIGHT - SEMANTICS

To explore the potential utility of the semantic and height information included in the scene map, we conducted a small experiment to evaluate how replacing binary obstacle maps with 3D height maps and semantic maps affects performance. Table11 presents the results of the global path prediction task using the UNet+A* model. Each model was trained and tested on LocoVR with binary maps, height maps, and semantic maps, over three trials. As shown in the table, the models trained with height and semantic maps clearly outperformed those trained with binary maps. Although we do not yet have a detailed analysis of these findings, they potentially suggest that human trajectories could be influenced by object attributes inferred from height and semantic information. For instance, participants might unconsciously maintain a distance from movable objects, such as chairs or doors, or adjust their trajectories based on the visual clearance provided by different object types. For example, walls, kitchen counters, and low tables offer varying degrees of vision clearance, with lower clearance potentially exerting subtle psychological pressure on trajectory planning. A detailed analysis on influence of variational scene information on the human trajectories could provide valuable insights from the perspectives of cognitive and behavioral sciences.

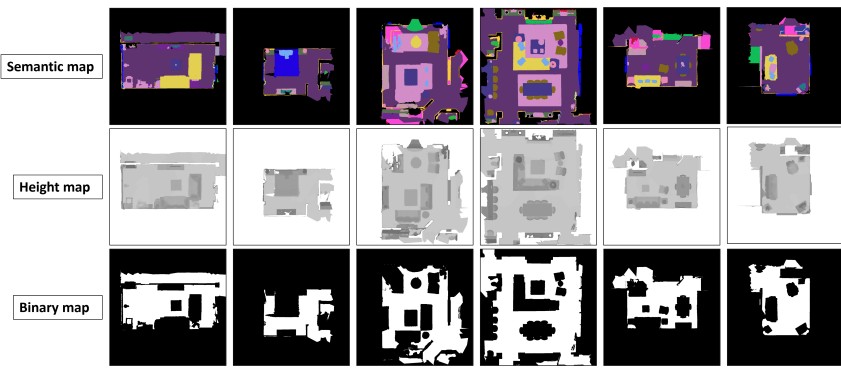

Figure 6: Examples of binary (obstacle) map, height map, and semantic map

Table 11: Global path prediction with binary/semantic/height maps - ADE (Average Displacement Error) between predicted and ground-truth time-series trajectories. The table reports averaged performance in all of the trajectories over three trials $\pm$ SD

| Method | $0m \leq d \leq 3m$ | $3m \leq d \leq 6m$ | $6m \leq d$ |
|---|---|---|---|
| Binary map | $0.138_{\pm 0.0006}$ | $0.183_{\pm 0.0024}$ | $0.286_{\pm 0.0113}$ |
| Semantic map | $0.137_{\pm 0.0004}$ | $1.70_{\pm 0.0046}$ | $0.216_{\pm 0.0278}$ |
| Height map | $\mathbf{0.136}_{\pm 0.0011}$ | $\mathbf{0.165}_{\pm 0.0068}$ | $\mathbf{0.201}_{\pm 0.0219}$ |

# E   SOCIAL MOTION BEHAVIOR

Typically, a person's trajectory is influenced by the movements of others who are close by, as people naturally consider how their motion behavior might impact others in close proximity and modify their own motion behavior to accommodate others. Even when people are a bit farther apart, social motion behaviors can still occur, such as navigating around each other to respect personal spaceBurgoon & Hubbard (2005) or choosing a less direct route to avoid collision.

Our LocoVR dataset offers a unique perspective on social navigation dynamics within home environments by focusing on how people navigate shared spaces. Nearly half of the trajectories in our dataset involve individuals coming within 1.5 meters of each other (as seen in Figure.16), capturing a range of direct and indirect interpersonal space interactions. By analyzing the overlap of personal space volumes, we can identify moments of close proximity that require mutual awareness and behavioral adjustments. These interactions, while not as overt as handshakes or object exchanges, reveal subtle yet crucial aspects of cohabitation. They showcase how individuals modify their movements in response to another's presence – slowing down, altering paths, maintaining respectful distances, or yielding a path. This focus on spatial negotiation provides valuable insights into the unspoken choreography of daily life that occurs when sharing living quarters.

Figure7 illustrate three types of social navigation dynamics between two individuals (p1 and p2) in the real-world test dataset LocoReal. Each figure exemplifies different types of social motion behaviors, including maintaining social distance, stepping aside to allow others to pass, and choosing which side of an object to pass on to avoid crossing paths. Also, We demonstrate our model's capability to perform social navigation by considering the trajectories of others. Each row shows the results with and without using P2's past trajectories as input, while each column represents a different sample scene. We used the model A*+U-Net(LocoVR), presented in Section4.3.1, throughout this experiment.

The dark green, light green, and red lines represent p1's past trajectory, true future path, and the predicted path by our model, respectively. The orange circle marks p1's goal position. Additionally, the blue line and light blue arrow indicate P2's past trajectory and direction of movement, respectively.

Scene 1 to 3 depict scenes where p1 and p2 are about to cross paths. The groundtruth future path (light green) shows that p1 maintains an appropriate social distance from p2's path. Focusing the predicted future trajectories by our model (red), the upper row of the figure shows that the model predicts a path that overlaps with p2's heading direction since the model is not able to obtain p2's movement at all. In contrast, the lower row demonstrates that the model generates a path that maintains a certain distance from p2's heading direction, similar to the ground truth. This indicates that the model has effectively learned social motion behavior from LocoVR dataset.

Scene 4 to 6 illustrate situations where p1 chooses longer paths to avoid interference with p2. The ground truth future path of p1 demonstrates a choice that avoids potential proximity to p2 by selecting a route where p2 is absent. In the upper row, where the model does not consider P2's trajectory, the predicted path generally follows the geometrically efficient route to the goal, which could potentially overlap with p2's movement area. In contrast, the lower row shows that the model prioritizes avoiding potential proximity to p2, indicating that it has learned to account for social motion behavior.

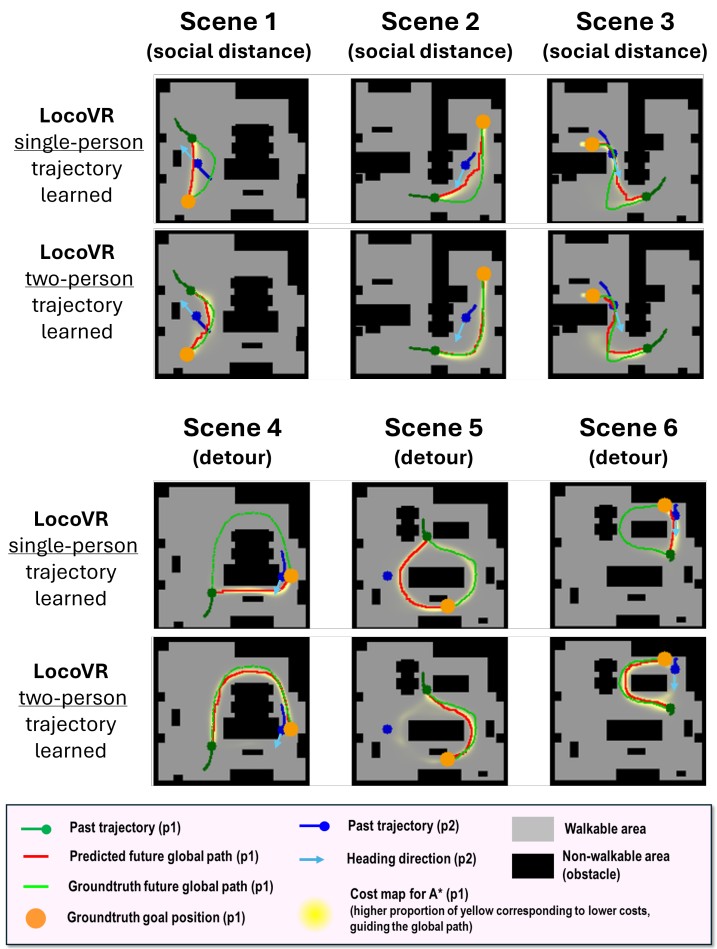

Figure 7: Social motion behavior

# F   VR SYSTEM FOR DATA COLLECTION

## F.1   SYSTEM STRUCTURE

The system receives real-time tracking data from the motion trackers worn by each participant. This data is used to update the avatars in the virtual environment so they accurately reflect the poses and movements of the participants. Each virtual avatar is represented by a SMPL mesh (Loper et al., 2015), calibrated to match the proportions of each person's body using FINAL-IK (Root Motion, 2024). To encourage movement and social motion related behaviors between the two participants, the system generates a goal object that each person needs to reach as part of the data collection task. This gives the participants a reason to move around and interact with each other in the virtual space, resulting in social behaviors, such as waiting for someone to pass before proceeding or backtracking for an oncoming person if the path is too narrow to allow them to cross.

## F.2   VR HARDWARE

Using the HTC VIVE system (VIVE, 2023), we track the movements of two people as they explore a virtual space. Each person wears a VR headset, holds a controller in each hand, and has three motion trackers (VIVE pucks) on their body - two on the ankles and one on the torso, for a total of six tracked points. This setup allows us to capture the full range of body movements as the participants interact with the virtual environment and with each other. The HTC VIVE's outside-in tracking system uses the six tracked points on each person's body to calculate their absolute pose and position with a high

degree of accuracy. With a tracking frequency of 90 Hz and an accuracy within a few millimeters (Holzwarth et al., 2021), the system can track small movements and gestures in real-time to provide a highly responsive and immersive experience, thereby eliciting natural walking and social behaviors necessary for collecting realistic data.

### F.3 3D VIRTUAL ROOM DATA

We use the Habitat-matterport 3D dataset (Ramakrishnan et al., 2021) to create the virtual scenes. The dataset includes more than 1K 3D indoor spaces, which are captures of actual rooms using the Matterport 3D scanner (Matterport, 2023). Some of the scenes have semantic information added that Matterport has manually labeled. We used 131 scenes with full 3D scene geometry and semantic labels to collect our data.

### F.4 ALIGNMENT OF VR AND THE REAL SPACE

The data collection was conducted in a VR lab (10m x 10m), which was larger than every virtual room we used. Firstly, we aligned the centers of the virtual and the physical rooms so that the virtual room was totally contained in the physical room. During data collection, goal positions were controlled to appear in the predefined virtual room to make the participants walk safely without getting close to the physical walls. If a participant got close to the physical walls, a virtual guardian appeared, indicating to the participant that they were too close to the boundary of the virtual space. This is a built-in safety feature of the VR headset we used.

### F.5 AVATAR CALIBRATION

We employed inverse kinematics software, FINAL-IK to reconstruct avatar motion from tracked body motions on the head/waist/hands/feet. Due to the performance of FINAL-IK, the avatar motions may occasionally display unnatural joint movements. However, these slight inaccuracies in body motion do not impact the contribution of our experiment, as our focus is on room-scale human dynamics rather than fine-grained body movements. Our dataset currently includes raw data from sparse motion trackers, which is highly accurate (within a few millimeters). For users requiring precise avatar motion, applying state-of-the-art IK algorithms to the raw tracker data would reconstruct more accurate avatar movements than those displayed in our video.

## G LOCOREAL: A DATASET FOR TESTING IN THE REAL WORLD

Although LocoVR is collected in highly realistic virtual environments and useful for learning human trajectory considering the surrounding environment, it is a general concern that there might be a difference in human perception between the physical and virtual space that results in performance degradation when transferring from the virtual to the real world. To address the concern, we built LocoReal, a human trajectory dataset in the physical space, which can be used as test data to show that the model trained with LocoVR can be utilized in the real environment.

Collecting real-world human trajectory data was done in an empty room in a campus building. Two participants walked to conduct a task in the room where several pieces of furniture were placed, and their 3D motions and trajectories were captured by a motion capture system. The experiment was conducted in 4 different layouts with 5 participants, resulting in 450 collected trajectories. Figure 8 illustrates the binary maps of the 4 scenes we collected in LocoReal.

## H DATA AUGMENTATION

All the training data are augmented by horizontal flipping and rotation with 90, 180, and 270 degrees, increasing the number of training data by eight-fold. We also augmented the data in a time-series direction to obtain different sets of past trajectories and ground-truth future trajectories. We define a fixed length time window and slide it with a certain interval to obtain sets of augmented trajectories. Fig 9 shows an overview of the data augmentation strategy. Each task has a different strategy in data augmentation.

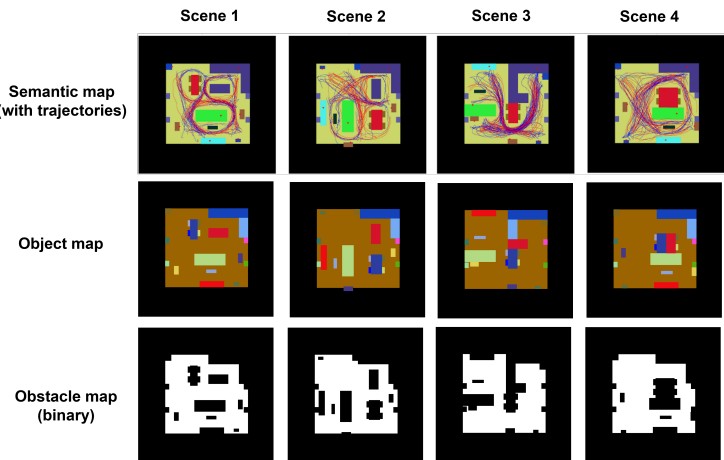

Figure 8: Maps data of LocoReal dataset. We collected the dataset with 5 participants performing tasks in the shown 4 different layouts of a physical room.

For the global path prediction, the time window with a length of 10.0s (1.0s for past trajectory and 9.0s for future trajectory) slides with an interval of 0.67s to extract trajectory segments from the raw trajectory. The duration of 10.0s is chosen to encompass the majority of trajectories in the dataset (as shown in Figure 13), while a time step of 0.67 seconds is sufficient to capture human movement at speeds of up to 3.0 m/s. A part of the trajectory segment that is out of the raw trajectory is padded with the last value observed in the raw trajectory.

For the trajectory prediction, the time window with a length of 4.0s (1.0s for past trajectory and 3.0s for future trajectory) slides with an interval of 0.67s to extract trajectory segments from the raw trajectory. We assume that 3.0s is an appropriate time range for future trajectory prediction, as this task is non-goal-conditioned and affected by rapidly increasing positional uncertainty over time. Note that all the trajectory segments are within the raw trajectory because the task focuses on trajectory prediction in local areas on the way to the goal.

For the goal prediction, the time window with a length of 6.0s (for past trajectory) slides with an interval of 0.67s to extract trajectory segments from the raw trajectory. We set 6.0 seconds as the maximum length for past observations, as the primary objective is to predict the goal before it is reached, making long observation periods unnecessary for evaluating goal prediction performance. A part of the trajectory segment that is out of the raw trajectory is padded with the initial value observed in the raw trajectory.

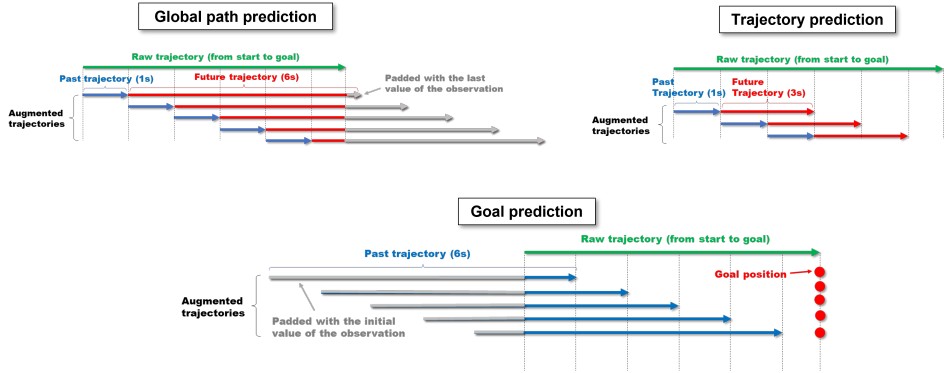

Figure 9: Data augmentations in the time direction

# I LocoVR dataset statistics

In this section, we describe the statistics of the trajectory data contained in LocoVR. We collected trajectory data from 32 participants in total, resulting in 7071 trajectory sequences after data pre-processing. Since we collected trajectories from two participants simultaneously, each participant's trajectory was counted separately. We removed short trajectories (less than 2m or 2s) and poor motion tracking data in the data preprocessing phase.

## I.1 Statistics on the data

Figure 10 shows the number of trajectories collected in each scene. The average number and standard deviation over 131 scenes are 54.0 and 32.0, respectively. The number of trajectories differs across scenes, resulting from the following factors. We collected a large number of trajectories in scenes where human interactions occur frequently (e.g., paths with a bottleneck). Also, the number of trajectories is affected by the speed preferences of the participants. Given the same amount of time, participants walking fast gave us more trajectories than participants who walked slowly. Further, the stability of the motion-tracking performance also affected the number of trajectories since trajectories with large tracking errors are removed in the data preprocessing.

Figure 11 Path efficiency is a metric used to evaluate the complexity of a trajectory, defined as the ratio of the straight-line distance to the goal to the actual traveled distance. The mean path efficiency in LocoVR is 0.81, comparable to that reported in THOR-MAGNI (Schreiter et al., 2024). Notably, the data in THOR-MAGNI were collected in a single, controlled experimental setting designed to emphasize the influence of surrounding objects or other pedestrians. In contrast, our dataset was collected in diverse, naturally occurring home environments that were not controlled by experimenters.

Figure 12 shows the distance distributions of the trajectories. The figure shows that more than half of the trajectories are longer than 4m. Since virtual rooms are usually smaller than 7m by 7m, the distribution is reasonable to assume daily movements in a room. Figure 13 shows the travel time distribution of the trajectories. It shows that more than half of the trajectories are longer than 5s, which would be enough to learn locomotion in a single room.

Figure 14 shows the speed distribution of the trajectories. The mean value is around 0.8, indicating the walking speed is relatively low compared to the open outdoor space since the scale of home environments is smaller.

Figure 15 depicts the number of peaks observed in the speed history of individual trajectories. It shows that more than half of the trajectories exhibit two or more peaks, indicating that participants frequently change their speed on their way to the goal. This behavior is likely influenced by interactions with other participants or the narrow and complex indoor environment.

Figure 16 shows the minimum distance between two participants in each trajectory. It is shown that approximately 25% of the trajectories are within 1m of the other participant, and more than 70% are within 2m (See Figure 18). It indicates that many of the trajectories could be influenced by the trajectories of the other participants when they are in close proximity, as people typically consider how their behaviors might affect others when they are located close to other people. In the rest of the cases where the participants were at least 2m away from each other, there could still be social motion behaviors that involve passing through each other at a distance to respect other people's personal space or taking a less direct route to the goal to avoid the risk of physical conflict with the other.

Figure 17 illustrates the distribution of closing speeds between participants. The relatively high closing speeds, considering the room scale, suggest that participants need to remain attentive to the movements of others.

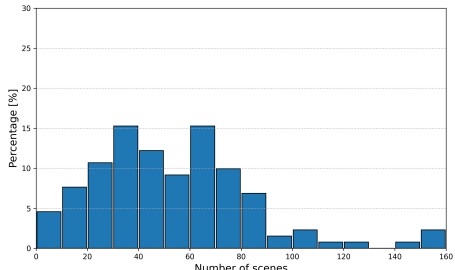

Figure 10: Number of trajectories in each scene

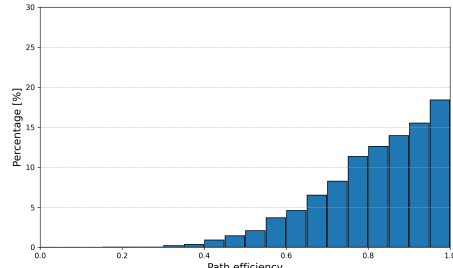

Figure 11: Path efficiency (straight-line distance to the goal/traveled distance) in each trajectory.

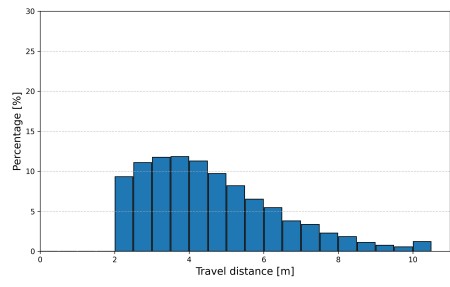

Figure 12: Travel distance in each trajectory

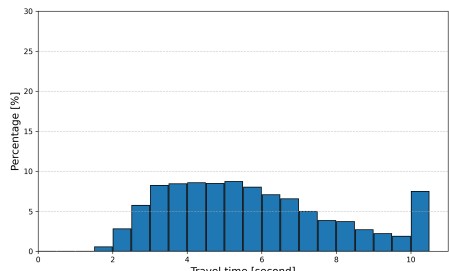

Figure 13: Travel time in each trajectory

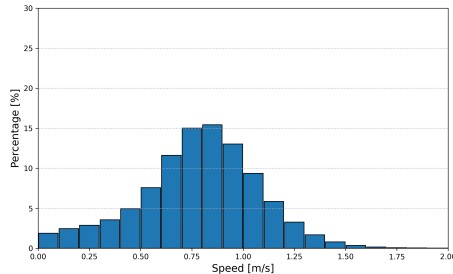

Figure 14: Speed of participants.

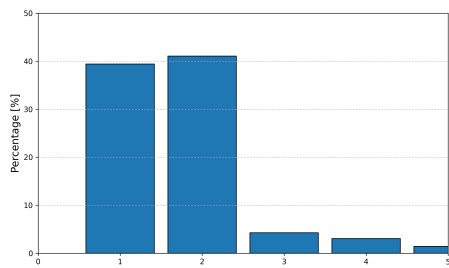

Figure 15: Number of peaks in speed history in each trajectory

## I.2 STATISTICS ON THE PARTICIPANTS

There were 32 participants in total, comprising 21 males and 11 females, with ages ranging from 18 to 42. From this pool, pairs were formed to conduct 25 experiments, each involving a unique pair (Table12). The experiments included various combinations of male-male, female-female, and male-female pairs, as well as pairs of friends and nonfriends, as shown in the Table13. As reactions between pairs in close proximity could influenced by attributes and interpersonal relationships, further data analysis may provide new insights into the relationship between these attributes, relationships, and behavioral patterns. It could be an intriguing study from the perspective of cognitive and behavioral sciences.

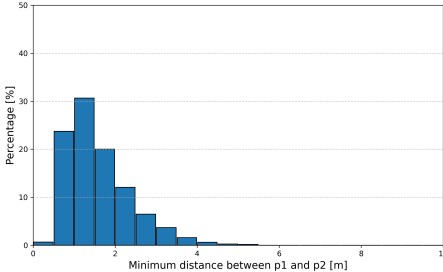

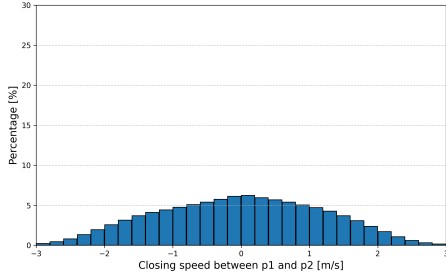

Figure 16: Minimum distance between partici-
pants in each trajectory.

Figure 17: Closing speed between participants.

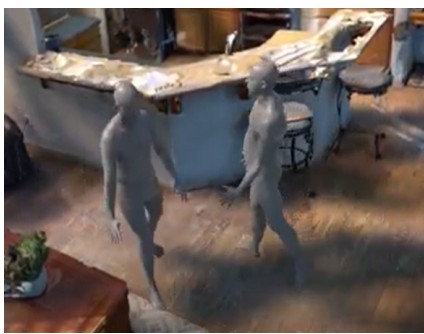

Figure 18: Scene of participants approaching each other

Table 12: Participants demographics

| Age | Number of participants | |
|---|---|---|
| | Male | Female |
| Under 20 | 5 | 3 |
| 20 to 29 | 15 | 8 |
| Over 30 | 1 | 0 |

Table 13: Diversity of pairs

| Relationship | Number of participants | | |
|---|---|---|---|
| | Male − Male | Female − Female | Male − Female |
| Friends | 2 | 2 | 5 |
| Nonfriends | 9 | 1 | 6 |

## I.3   WHY TWO PERSON ? - MOTIVATION OF TWO-PERSON EXPERIMENT

In contrast to conventional studies that focus on crowd dynamics in open public spaces, our research emphasizes room-scale social motion behaviors between two individuals, particularly in confined indoor spaces. For example, interpersonal behaviors such as proxemics, trajectory negotiation, and mutual space adaptation are affected and induced by the narrow geometry. We consider this is a fundamental study on the room-scale social motion behaviors, and believe this focus does not limit the dataset's utility. Researchers can build upon it to study two-person interactions in isolation or as a basis for modeling interactions in more complex, multi-person environments. In addition, more than two-person interaction could be considered as a rare scenario in home environments. For instance, recent census data indicates that in the US, 60% of households consist of two people or fewer, and 80% consist of three people or fewer. Also, indoor home spaces are generally smaller than open public spaces, and activities in such settings are often more individualized. As a result, residents

typically move independently rather than engaging in collaborative or coordinated movements within the same space. This relevance underscores the practical utility of our dataset for studying interaction dynamics that are directly applicable to real-world scenarios.

## J    INFLUENCE OF GAP BETWEEN VR AND THE REAL

We consider the virtual avatar an effective tool for addressing the VR/real-world gap. Additionally, we introduced a filter to exclude data exhibiting inappropriate behaviors. In the following sections, we discuss the impact of this gap.

(1) In our experiment, participants were aware that the virtual avatars were synchronized with real humans sharing the same physical space. Also, the avatar enables participants to percept relative position between their body and surrounding objects. These awareness discouraged socially or physically inappropriate behavior, mitigating the potential impact of the VR/real gap, as demonstrated in recent studies on VR locomotionYun et al. (2024); Simeone et al. (2017). In addition, we have introduced a filter to detect instances of users passing through virtual objects to remove such data from the dataset.

(2) Our evaluation used locomotion data collected in physical spaces as test data. Models trained on the LocoVR dataset outperformed those trained on other physically collected datasets (GIMO/THOR-MAGNI), demonstrating that VR-collected data is effective when applied to real-world scenarios.

For the futurework, more expressive avatar to facilitate non-verbal communication during walk, or user interaction cues such as haptic devices could further reduce the gap between VR/Real. We anticipate that future advancements in VR technologies will further contribute to bridging this gap.

