# OpenReview forum: "LocoVR: Multiuser Indoor Locomotion Dataset in Virtual Reality"
_ICLR.cc/2025/Conference — ICLR 2025 Poster_

### Official Review · Reviewer_PhCB · 2024-10-31

**Soundness:** 3
**Presentation:** 3
**Contribution:** 3
**Rating:** 6
**Confidence:** 4

**Summary:**

This paper focuses on understanding human locomotion behaviors in indoor environments, with the specific scenario of two persons walking two separate goals in an indoor room. The main contribution is a new dataset LocoVR capturing sequences of two persons walking to their goal locations in 3D indoor rooms. To overcome the high cost of capturing locomotion in physical scenes, this paper proposes a VR-based solution where the subjects wear a VR device and navigate through a virtual 3D room displayed in VR. The human locomotion trajectories are tracked using VR devices. This VR capture solution scales up locomotion capture to 130+ scenes. The authors then conducted experiments validating that the collected LocoVR dataset outperforms existing datasets in three tasks: global path prediction, trajectory prediction, and goal prediction.

**Strengths:**

1. The proposed VR capture solution eliminates the high cost of physically setting up indoor scenes and capturing human movements, which facilitates scaling up locomotion capture to many more scenes compared to previous datasets.

2. This paper captures two-person goal-reaching motions which include the social navigation behaviors such as adjusting the path to respect personal boundaries or side steps to give way to another person. Such social navigation behavior is not covered in most previous datasets and is important for understanding multi-human social navigation and potential human-robot interactions.

3. Experiments on three navigation-related tasks show that models trained on the LocoVR dataset consistently outperform models trained on existing datasets when tested on a real-world two-person locomotion test set.

**Weaknesses:**

1. Although this paper claims to provide full body pose data (L20), the human motion capture is far from realistic according to the supplementary video (0:00-0:30). If aiming for capturing full body poses, it may be necessary to change from the HTC VIVE tracking to marker-based motion capture as in CIRCLE (Araujo et al., 2023). With the current presented results, I recommend removing the claims of full body pose data since all experiments only use trajectory data.

2. The VR capture system is limited to simple behaviors assuming a flat floor scene and no contact-based close object interactions. It can only capture locomotion or reaching behaviors as in CIRCLE (Araujo et al., 2023). The VR capture system can not work for behaviors like lying on a sofa or walking up stairs. When the humans try to do such interactions in VR, they are actually interacting with air in the real world and will fall. This virutal-real inconsistency can also cause the subjects to slightly walk into obstacles as discussed in paper.

3. This paper only focus on a very simple social navigation scenario of two persons avoiding each other. However, the social navigation behaviors can be much more complex. For example, humans do not only avoid each other but also collaborates and coordinates, consider the cases where one person is leading the way and other persons follow the leading one, and two persons walk to each other to talk. It is also necessary to include social scenarios with more than two persons.

**Questions:**

1. In appendix H, why are time windows and intervals set as the presented numbers? Are there any motivation or empirical study?

2.L860, figures should be tables?

---

> ### Author Response · Authors · 2024-11-22
> **Response to Reviewer PhCB (1/2)**
>
> We sincerely thank the reviewer for their valuable feedback, which has greatly enhanced the clarity and quality of our paper.
> In response to the reviewer's concerns, we have carefully addressed them as outlined below.
>
> ---
>
> > **R1:
> Although this paper claims to provide full body pose data (L20), the human motion capture is far from realistic according to the supplementary video (0:00-0:30). If aiming for capturing full body poses, it may be necessary to change from the HTC VIVE tracking to marker-based motion capture as in CIRCLE (Araujo et al., 2023). With the current presented results, I recommend removing the claims of full body pose data since all experiments only use trajectory data.**
>
> **A1:**
>
> **Full-body pose is auxiliary information but not our main focus:**
> We incorporated the motion capture system primarily to visualize avatar motions, allowing participants to recognize the movements of others. Consequently, including full-body motion data in the dataset is not our primary focus. In the main paper, we mentioned full-body motion as auxiliary information included in the dataset, noting that head pose (yaw direction) from the raw motion tracking data was incorporated to maximize performance in our evaluation tasks. The head pose offers valuable insights for inferring human intentions, which facilitates more accurate future predictions, as demonstrated in the ablation study (Appendix C).
>
> **Inaccuracies in avatar motions:**
> As the reviewer noted, avatar motions may occasionally display unnatural joint movements. This issue arises from the performance of the inverse kinematics (IK) software (FINAL-IK), which reconstructs avatar motion using sparse motion trackers placed on the body (head, waist, hands, and feet).
> However, these slight inaccuracies in body motion do not impact the contribution of our experiment, as our focus is on room-scale human dynamics rather than fine-grained body movements.
> Our dataset currently includes raw data from sparse motion trackers, which is highly accurate (within a few millimeters). For users requiring precise avatar motion, applying state-of-the-art IK algorithms to the raw tracker data would reconstruct more accurate avatar movements than those displayed in our video.
>
> ---
>
> > **R2:
> The VR capture system is limited to simple behaviors assuming a flat floor scene and no contact-based close object interactions. It can only capture locomotion or reaching behaviors as in CIRCLE (Araujo et al., 2023). The VR capture system can not work for behaviors like lying on a sofa or walking up stairs. When the humans try to do such interactions in VR, they are actually interacting with air in the real world and will fall. This virutal-real inconsistency can also cause the subjects to slightly walk into obstacles as discussed in paper.**
>
> **A2:**
> We agree with the reviewer that there is a gap between VR and reality, and it could influence the human behavior. However, we believe this gap has minimal influence on our locomotion experiment and does not make impact on the overall contribution of our dataset for the following reasons:
>
> (1) In our experiment, participants were aware that the virtual avatars were synchronized with real humans sharing the same physical space. Also, the avatar enables participants to percept relative position between their bodies and surrounding objects. These awareness discouraged socially or physically inappropriate behavior, mitigating the potential impact of the VR/real gap, as demonstrated in a recent study on VR locomotion[1][2]. In addition, we have introduced a filter to detect instances of users passing through virtual objects to remove such data from the dataset.
>
> (2) Our evaluation used locomotion data collected in physical spaces as test data. Models trained on the LocoVR dataset outperformed those trained on other physically collected datasets (GIMO/THOR-MAGNI), demonstrating that VR-collected data is effective when applied to real-world scenarios.(Tested on LocoReal: Main paper Section.4, tested on GIMO: Appendix D.1)
>
> We have included above discussion in the revised manuscript to clarify the influence of VR/Real gap issue. (Appendix.J, highlighted in blue)
>
> [1] H. Yun, Y. Watanabe, A. Yamada, "Exploring the Role of Expected Collision Feedback in Crowded Virtual Environments," Proc. IEEE Conf. Virtual Reality and 3D User Interfaces, 2024.
>
> [2] A. L. Simeone, I. Mavridou, and W. Powell, "Altering user movement behaviour in virtual environments," IEEE Transactions on Visualization and Computer Graphics, vol. 23, no. 4, pp. 1312–1321, 2017.

---

> > ### Comment · Reviewer_PhCB · 2024-11-22
> >
> > Thanks for the response and discussions.  Given the quality of the provided full-body pose data, I still recommend positioning the dataset as a trajectory dataset or clearly stating that the full-body poses are auxiliary, inaccurate estimations from sparse trackers.

---

> ### Author Response · Authors · 2024-11-22
> **Response to Reviewer PhCB (2/2)**
>
> ---
>
> > **R3:
> This paper only focus on a very simple social navigation scenario of two persons avoiding each other. However, the social navigation behaviors can be much more complex. For example, humans do not only avoid each other but also collaborates and coordinates, consider the cases where one person is leading the way and other persons follow the leading one, and two persons walk to each other to talk. It is also necessary to include social scenarios with more than two persons.**
>
> **A3:**
> We agree that variations in locomotion patterns involving more than two individuals may occur in real-world scenarios, particularly in open public spaces. However, our research focuses on private indoor settings where the number of pedestrians is typically very limited, and individuals generally move independently. We believe this focus does not diminish the contribution of our dataset, as scenarios in private indoor settings are both common and essential in real-world contexts, yet they have been largely overlooked by most existing datasets. Please see the following for a detailed discussion.
>
>   - **Our goal:**
> In contrast to conventional studies emphasizing crowd dynamics in open public spaces, our research primarily focuses on social motion behaviors in room-scale private settings. In such environments, individuals exhibit more individualized social behaviors constrained by narrow geometries, such as taking longer detours to avoid others, yielding paths, or maintaining social distance while passing. Given the lack of existing datasets addressing this specific problem setting, our goal is to provide a fundamental resource that is both targeted and extensible, serving as a stepping stone for future datasets that could scale up to include more individuals or different interaction settings.
>
>   - **Two-person navigation scenarios in real-world relevance:**
> In home environments, interactions involving more than two people are relatively uncommon. Recent census data indicates that 60% of households in the U.S. consist of two people or fewer. Even in households with three or more members, scenarios where more than two individuals navigate a space simultaneously are rare, as residents generally move independently rather than engaging in collaborative or coordinated movements within the private setting. Similarly, other private spaces such as small offices, clinics, or hotel rooms are common examples of everyday environments where two-person navigation is prevalent.
> This relevance underscores the utility of our dataset for studying interaction dynamics that are directly applicable to these contexts.
>
>   - **Potential utility of two-person navigation dataset:**
> Two-person interactions are foundational to understanding more complex multi-person dynamics, as they allow us to study detailed interpersonal behaviors such as proxemics, trajectory negotiation, and mutual space adaptation without the confounding variables introduced by larger groups.
> In the future, researchers can build upon our dataset to study two-person interactions in isolation or as a basis for modeling interactions in more complex, multi-person environments.
>
> We thank the reviewer for highlighting this point, and we included the discussion in the paper.
> (Appendix.I.3, highlighted in blue)
>
> ---
>
> > **R4:
> In appendix H, why are time windows and intervals set as the presented numbers? Are there any motivation or empirical study?**
>
> **A4:**
> We thank the reviewer for highlighting the motivation of the parameter settings, which allowed us to improve the clarity of the paper.
> The parameter settings are primarily motivated by the characteristics of each task and the statistics of the dataset. For detailed information, please refer to Appendix H in the revised main paper (Appendix.H, highlighted in blue).
>
> ---
>
> > **R5:
> L860, figures should be tables?**
>
> **A5:**
> We appreciate the reviewer for pointing out the mistake.
> We have modified the word, from "Figures" to "Tables". (Appendix.D, highlighted in blue)

---

> ### Author Response · Authors · 2024-11-22
> **Thank you for the response!**
>
> We appreciate the reviewer’s constructive suggestion. After careful consideration, we concluded that the revision better highlights the primary contribution of our dataset. The revised sentences in the manuscript are as follows:
>
> - Abstract (L20): Revised "full-body motion" to **"accurate trajectory data"**.
>
> - Introduction (L43): Revised "full-body motions" to **"trajectories"**.
>
> - Figure 1 caption (L89): Revised "full-body motion" to **"trajectory"**.
>
> - Section 3.1 Overview (L154): Revised "it includes full-body human poses, along with head orientation data in addition to trajectories" to **"it includes body tracker data on head/waist/hands/feet as auxiliary information"**.
>
> - Conclusion (L531): Revised "full-body motions" to **"accurate trajectory"**.
>
>
> All revisions have been highlighted in blue in the manuscript.
>
> ---

---

> ### Author Response · Authors · 2024-11-25
> **Welcome for further discussions!**
>
> Thank you once again for the insightful review. We believe the revision has strengthened the quality of our paper. If there is anything else we can clarify or elaborate on, please do not hesitate to let us know.

---

> ### Author Response · Authors · 2024-11-27
> **24 hours remaining before the paper revision deadline**
>
> Dear Reviewer PhCB,
>
> We hope this message finds you well. With 24 hours remaining before the revision deadline, we kindly request your feedback on the remaining concerns based on our responses. We understand the review process is time-consuming, but your feedback is invaluable in shaping the final outcome. Thank you again for your time and effort in reviewing our work.

---

### Official Review · Reviewer_Fpz9 · 2024-11-04

**Soundness:** 3
**Presentation:** 3
**Contribution:** 3
**Rating:** 6
**Confidence:** 3

**Summary:**

LocoVR is a virtual reality-based dataset aimed at improving the modeling of human locomotion in complex indoor environments. This dataset specifically focuses on multi-user indoor navigation, capturing over 7,000 two-person trajectories within more than 130 different home-like scenes. The main goal of the LocoVR dataset is to enhance the ability of AI systems, like home robots, to understand and predict human movement patterns that incorporate both spatial and social navigation dynamics.

**Strengths:**

By using VR, the dataset captures detailed spatial data and full-body motion in diverse home environments. The VR setup enables controlled capture of social navigation behaviors, such as maintaining personal space and avoiding collisions in shared spaces like entryways.

The dataset is well evaluated. The paper evaluates the dataset on three trajectory-based tasks—global path prediction, trajectory prediction, and goal prediction. The results show that models trained on LocoVR outperform those trained on other datasets, particularly in predicting realistic, socially aware navigation paths in complex environments.

I can see implications of this work not just for virtual agents, games etc, but also as we move to further robotic presence in our homes this type of dataset can help train their trajectories.

**Weaknesses:**

I m not sure about the Non-Verbal Social Cues: such as gaze direction or facial expressions—that influence social navigation.

The two agent approach perhaps limits also its scope in multi-user indoor scenarios common in real homes with multiple occupants.

**Questions:**

I would like future work to discuss how this type of work can be merged with other approaches like weighted interpolations to define trajectories of avatars indoors: https://www.microsoft.com/en-us/research/publication/avatarpilot-decoupling-one-to-one-motions-from-their-semantics-with-weighted-interpolations/

---

> ### Author Response · Authors · 2024-11-22
> **Response to Reviewer Fpz9**
>
> We sincerely thank the reviewer for their valuable feedback, which has greatly enhanced the clarity and quality of our paper.
> In response to the reviewer's concerns, we have carefully addressed them as outlined below.
>
> ---
>
> > **R1:
> I'm not sure about the Non-Verbal Social Cues: such as gaze direction or facial expressions—that influence social navigation.**
>
> **A1:**
> We consider the lower fidelity of the SMPL avatar as a potential factor contributing to the gap between VR and real-world scenarios, although its impact appears to be minor. For example, we often rely on observing others' gaze to predict their heading direction or use facial expressions to communicate when yielding a path. Incorporating more realistic and expressive avatars to enhance the integrity of the VR-based data collection framework remains an avenue for future work.
>
> ---
>
> > **R2:
> The two agent approach perhaps limits also its scope in multi-user indoor scenarios common in real homes with multiple occupants.**
>
> We agree that variations in locomotion patterns involving more than two individuals may occur in real-world scenarios, particularly in open public spaces. However, our research focuses on private indoor settings where the number of pedestrians is typically very limited, and individuals generally move independently. We believe this focus does not diminish the contribution of our dataset, as scenarios in private indoor settings are both common and essential in real-world contexts, yet they have been largely overlooked by most existing datasets. Please see the following for a detailed discussion.
>
> **A2:**
>   - **Our goal:**
> In contrast to conventional studies emphasizing crowd dynamics in open public spaces, our research primarily focuses on social motion behaviors in room-scale private settings. In such environments, individuals exhibit more individualized social behaviors constrained by narrow geometries, such as taking longer detours to avoid others, yielding paths, or maintaining social distance while passing.
> Given the lack of existing datasets addressing this specific problem setting, our goal is to provide a fundamental resource that is both targeted and extensible, serving as a stepping stone for future datasets that could scale up to include more individuals or different interaction settings.
>
>   - **Two-person navigation scenarios in real-world relevance:**
> In home environments, interactions involving more than two people are relatively uncommon. Recent census data indicates that 60% of households in the U.S. consist of two people or fewer. Even in households with three or more members, scenarios where more than two individuals navigate a space simultaneously are rare, as residents generally move independently rather than engaging in collaborative or coordinated movements within the private setting. Similarly, other private spaces such as small offices, clinics, or hotel rooms are common examples of everyday environments where two-person navigation is prevalent.
> This relevance underscores the utility of our dataset for studying interaction dynamics that are directly applicable to these contexts.
>
>   - **Potential utility of two-person navigation dataset:**
> Two-person interactions are foundational to understanding more complex multi-person dynamics, as they allow us to study detailed interpersonal behaviors such as proxemics, trajectory negotiation, and mutual space adaptation without the confounding variables introduced by larger groups.
> In the future, researchers can build upon our dataset to study two-person interactions in isolation or as a basis for modeling interactions in more complex, multi-person environments.
>
>
> We thank the reviewer for highlighting this point, and we ensure that our intentions and the real-world relevance of two-person interactions have been included in the paper.
> (Appendix.I.3, highlighted in blue)
>
> ---
>
> > **R3:
> I would like future work to discuss how this type of work can be merged with other approaches like weighted interpolations to define trajectories of avatars indoors: https://www.microsoft.com/en-us/research/publication/avatarpilot-decoupling-one-to-one-motions-from-their-semantics-with-weighted-interpolations/**
>
> **A3:**
> We thank the reviewer for providing insightful suggestions to enhance the clarity of our contribution. We have added the discussion of future applications, referring to the paper the reviewer suggested. (Section.5, highlighted in blue)
>
>
> ---
>
> **[Final Note]** Thank you once again for the insightful review. We believe the revision has strengthened the quality of our paper. If there is anything else we can clarify or elaborate on, please do not hesitate to let us know.

---

> ### Author Response · Authors · 2024-11-25
> **Discussion period approaches its conclusion in the coming days**
>
> **We have done our utmost to address the concerns raised and improve our work based on your valuable comments.**
>
> **If you have any additional questions or points of clarification, we would be delighted to engage in further discussion to ensure that all your concerns are thoroughly resolved.**
>
> **Thank you once again for your invaluable contributions, and we look forward to hearing from you.**

---

> ### Author Response · Authors · 2024-11-27
> **24 hours remaining before the paper revision deadline**
>
> Dear Reviewer Fpz9,
>
> We hope this message finds you well. With 24 hours remaining before the revision deadline, we kindly request your feedback on the remaining concerns based on our responses. We understand the review process is time-consuming, but your feedback is invaluable in shaping the final outcome. Thank you again for your time and effort in reviewing our work.

---

### Official Review · Reviewer_1K21 · 2024-11-04

**Soundness:** 4
**Presentation:** 4
**Contribution:** 4
**Rating:** 8
**Confidence:** 4

**Summary:**

1) LocoVR introduces a large dataset with 7000 two-person interactions across 130 diverse indoor environments in VR -- includes full body pose data and spatial information.
3) LocoVR improves model performance in 3 indoor tasks including human trajectories and predicting socially aware navigation patterns in home environments.

**Strengths:**

1) LocoVR’s large, indoor locomotion dataset records two-person interactions across 130 diverse indoor environments
2) Introduction of motion proxemics in a large-scale dataset which is useful for downstream tasks such as studying human-human interactions and potentially human-robot interactions
3) Rigorously quantitatively evaluated against strong baselines; baseline configurations and settings are fair and well-documented
4) Data of two-person trajectories is useful to study motion proxemics

The authors introduce an exciting/ relevant problem. The problem is well-motivated while the execution and evaluations are strong. I see many potential downstream applications and I believe that this will make a huge impact in the robotics field.

**Weaknesses:**

1) Potential Overfitting to VR-Specific Biases:
I am curious what the authors have done to further minimize the gap between real-world scenes vs VR scenes. Are there any obstacle perception features e.g., vibrational feedback when participants bump into objects in the scene?

2) *Qualitative results: ‘as the trajectory progresses, the probability distribution of the goal area narrows down near the true goal object’*
I think that it is reasonable to assume that humans narrow the probability distribution in LocoVR closer to the true goal object. However, I would like to understand exactly why this observation arises from LocoVR’s dataset. Is it merely the size of the dataset or is it the difference in data collected in LocoVR vs the other datasets? Are the authors claiming that this is a strength as it mimics the probability distribution of human trajectories better? If so, I would like to see analysis of this phenomenon in the different datasets.

3) Cultural/ societal biases of motion proxemics:
Since motion proxemics is influenced by cultural norms, authors should show aggregated user demographics. It would also be interesting to see motion proxemics based on the aggregated clusters of people.

**Questions:**

See weaknesses.

---

> ### Author Response · Authors · 2024-11-22
> **Response to Reviewer 1K21 (1/2)**
>
> We sincerely thank the reviewer for their valuable feedback, which has greatly enhanced the clarity and quality of our paper. In response to the reviewer's concerns, we have carefully addressed them as outlined below.
>
> ---
>
> > **R1:
> Potential Overfitting to VR-Specific Biases: I am curious what the authors have done to further minimize the gap between real-world scenes vs VR scenes. Are there any obstacle perception features e.g., vibrational feedback when participants bump into objects in the scene?**
>
> **A1:**
> We consider the virtual avatar as an effective tool to mitigate the VR/Real gap issue. In addition, we introduced a filter to remove data with inappropriate behaviors. In the following, we would like to discuss the influence of the gap issue.
>
> (1) In our experiment, participants were aware that the virtual avatars were synchronized with real humans sharing the same physical space. Also, the avatar enables participants to percept relative position between their body and surrounding objects. These awareness discouraged socially or physically inappropriate behavior, mitigating the potential impact of the VR/real gap, as demonstrated in a recent study on VR locomotion[1][2]. In addition, we have introduced a filter to detect instances of users passing through virtual objects to remove such data from the dataset.
>
> (2) Our evaluation used locomotion data collected in physical spaces as test data. Models trained on the LocoVR dataset outperformed those trained on other physically collected datasets (GIMO/THOR-MAGNI), demonstrating that VR-collected data is effective when applied to real-world scenarios.
>
> We also agree with the reviewer that user interaction cues such as haptic devices could reduce the gap between VR/Real. We anticipate that future advancements in VR user interfaces (VR-UI) will further contribute to bridging this gap.
>
> We have included above discussion in the revised manuscript to clarify the influence of VR/Real gap issue. (Appendix.J, highlighted in blue)
>
> [1] H. Yun, Y. Watanabe, A. Yamada, "Exploring the Role of Expected Collision Feedback in Crowded Virtual Environments," Proc. IEEE Conf. Virtual Reality and 3D User Interfaces, 2024.
>
> [2] A. L. Simeone, I. Mavridou, and W. Powell, "Altering user movement behaviour in virtual environments," IEEE Transactions on Visualization and Computer Graphics, vol. 23, no. 4, pp. 1312–1321, 2017.
>
> ---
>
> > **R2:
> Qualitative results: ‘as the trajectory progresses, the probability distribution of the goal area narrows down near the true goal object’ I think that it is reasonable to assume that humans narrow the probability distribution in LocoVR closer to the true goal object. However, I would like to understand exactly why this observation arises from LocoVR’s dataset. Is it merely the size of the dataset or is it the difference in data collected in LocoVR vs the other datasets? Are the authors claiming that this is a strength as it mimics the probability distribution of human trajectories better? If so, I would like to see analysis of this phenomenon in the different datasets.**
>
> **A2:**
> The improved performance in goal prediction using LocoVR compared to other datasets can be attributed to two key factors: (1) the dataset’s large size and diverse range of scenes, and (2) the complexity of those scenes. The extensive variety of human trajectories captured in diverse and complex environments helps the model learn the relationship between trajectories, scene layouts, and goal positions. This enables robust performance even in unseen home environments.
>
> To validate this, we evaluated the model trained on LocoVR using GIMO as the test dataset (Appendix.D.1, Table.10). The results consistently show that the model trained on LocoVR outperforms models trained on other datasets even in unseen scenes. It highlights LocoVR’s ability to enhance the model’s generalization by providing rich and diverse scene and trajectory data.

---

> ### Author Response · Authors · 2024-11-22
> **Response to Reviewer 1K21(2/2)**
>
> ---
>
> > **R3:
> Cultural/ societal biases of motion proxemics: Since motion proxemics is influenced by cultural norms, authors should show aggregated user demographics. It would also be interesting to see motion proxemics based on the aggregated clusters of people.**
>
> **A3:**
> There were 32 participants in total, comprising 21 males and 11 females, with ages ranging from 18 to 42. From this pool, pairs were formed to conduct 25 experiments, each involving a unique pair (Table.1).
> The experiments included various combinations of male-male, female-female, and male-female pairs, as well as pairs of friends and nonfriends, as shown in the table.2.
>
> As the reviewer pointed out, reactions between pairs in close proximity are influenced by attributes and interpersonal relationships. Further data analysis may provide new insights into the relationship between these attributes, relationships, and behavioral patterns. It could be an intriguing study from the perspective of cognitive and behavioral sciences. We appreciate the reviewer for suggesting the valuable insight, and have added the information shown in this reply in the revised manuscript. (Appendix.I.2, highlighted in blue)
>
> Table.1: User demographics (categorized by gender and age)
> |     Age    |   Male   |  Female  |
> |------------|----------|----------|
> |  under 20  |     5    |     3    |
> |  20 to 29  |     15   |     8    |
> |   over 30  |     1    |     0    |
>
>
> Table.2: Diversity of pairs (categorized by gender and relationship)
> |              |  Male-Male  | Female-Female | Male-Female |
> |--------------|-------------|---------------|-------------|
> |    Friends   |       2     |       2       |      5      |
> |  Non-friends |       9     |       1       |      6      |
>
>
>
>
> ---
>
> **[Final Note]** Thank you once again for the insightful review. We believe the revision has strengthened the quality of our paper. If there is anything else we can clarify or elaborate on, please do not hesitate to let us know.

---

> ### Author Response · Authors · 2024-11-25
> **Discussion period approaches its conclusion in the coming days**
>
> **We have done our utmost to address the concerns raised and improve our work based on your valuable comments.**
>
> **If you have any additional questions or points of clarification, we would be delighted to engage in further discussion to ensure that all your concerns are thoroughly resolved.**
>
> **Thank you once again for your invaluable contributions, and we look forward to hearing from you.**

---

> ### Author Response · Authors · 2024-11-27
> **24 hours remaining before the paper revision deadline**
>
> We hope this message finds you well. With 24 hours remaining before the revision deadline, we kindly request your feedback on the remaining concerns based on our responses. We understand the review process is time-consuming, but your feedback is invaluable in shaping the final outcome. Thank you again for your time and effort in reviewing our work.

---

> > ### Comment · Reviewer_1K21 · 2024-11-28
> >
> > Thank you for the response! A1 and A3 was well addressed. Additionally, thanks for the insight on relationships within the user study! A2 doesn't quite explain the phenomenon of the probability distribution narrowing down.
> >
> > I will maintain my score as I believe that this dataset will be useful to the community!

---

> > > ### Author Response · Authors · 2024-11-28
> > >
> > > Thank you once again for your time and effort in reviewing our work. We greatly appreciate your thoughtful and supportive comments in the review process.
> > >
> > > Regarding A2, we aim to provide a more comprehensive explanation to address the reviewer's question as follows. We hope this not only clarifies the reviewer's concerns but also benefits others who may have similar questions in the review.
> > >
> > > ---
> > > - **Additional explanation for A2:**
> > >
> > > To effectively predict goals based on human trajectories and scene layouts, it is crucial to accurately model the interdependent relationships between human trajectories, goal positions, and scene layouts. This necessitates datasets containing a large number of diverse combinations of human trajectories, goal positions, and scene layouts, collected across a wide variety of scenes to ensure robust generalization performance.
> > >
> > > However, existing datasets face challenges such as limited scene variation [1,2], lack of scene complexity [1], or insufficient numbers of trajectories [2], which often result in degraded performance when applied to unseen scenes. These limitations stem from the inherent difficulties of collecting data in real-world environments, where the process is time-consuming and constrained by accuracy issues due to blind spots and other observational limitations.
> > >
> > > In contrast, the LocoVR dataset contains a large number of trajectories spanning over 130 scenes, enabling improved generalization performance by effectively modeling the general relationships between human trajectories, goal positions, and scene layouts in indoor home environments.
> > >
> > > Our experiments demonstrate that models trained on LocoVR outperform those trained on other datasets[1,2] in goal prediction tasks, highlighting its advantages for achieving superior performance in diverse and unseen scenarios. (4.5.3 Table.4, Appendix D.1 Table.10)
> > >
> > > **Reference:**
> > >
> > > [1]Schreiter, Tim, et al. "THÖR-MAGNI: A large-scale indoor motion capture recording of human movement and robot interaction." The International Journal of Robotics Research (2024).
> > >
> > > [2]Zheng, Yang, et al. "Gimo: Gaze-informed human motion prediction in context." European Conference on Computer Vision. Cham: Springer Nature Switzerland, 2022.

---

### Official Review · Reviewer_6iZX · 2024-11-04

**Soundness:** 3
**Presentation:** 3
**Contribution:** 2
**Rating:** 6
**Confidence:** 3

**Summary:**

The paper introduces LocoVR, a virtual reality dataset that captures approximately 7,000 two-person trajectories across 130 indoor home scenes. The authors demonstrate the utility of LocoVR through three applications: global path prediction, trajectory prediction, and goal area prediction. The model trained on this dataset exhibits socially and geometrically aware navigation patterns within indoor scenes.

**Strengths:**

1. The extensive collection of two-person human trajectory (with motion capture) data in indoor scenes, is a valuable resource for the community.
2. Utilizing virtual reality for data collection is a promising approach, allowing more diverse 3D scenes when resources are limited.
3. The data and code will be released.

**Weaknesses:**

1. While using VR to collect human trajectory data is helpful, this paper would benefit from a discussion in the related works section about VR and human motion. For instance, referencing works like "QuestEnvSim: Environment-aware Simulated Motion Tracking from Sparse Data" in SIGGRAPH 2023 which uses VR for motion tracking and "Strategy and Skill Learning for Physics-based Table Tennis Animation" in SIGGRAPH 2024 which involves interaction between human and humanoid agents.
2. I notice authors utilize motion capture to provide whole body motion, and I wonder the reason to consider only experiments of path, trajectory and goal prediction. The occasionally unnatural motion observed in the video could be explained.
3. The use of A* baselines seems inappropriate for two-person interaction scenarios. I notice this dataset mainly focuses on obstacle avoidance. There appears to be a lack of interactive behaviors between the two persons. It may not be enough if two persons just operate independently and avoid the other person within the same space. I think it doesn't reflect scenarios often seen in real life. Can the authors provide more information on the distribution of action types within the dataset? Given this is a dataset paper, more statistics and descriptions would be beneficial.

**Questions:**

The paper seems to focus on locomotion. Without interactions like sitting on a sofa or standing up from a chair, does the goal prediction remain compelling?
With many figures presented in 2D planes, would a bird's eye view semantic map provide enough information for the prediction tasks? What's the importance of 3D geometry?

---

> ### Author Response · Authors · 2024-11-22
> **Response to Reviewer 6iZX (1/3)**
>
> We sincerely thank the reviewer for their valuable feedback, which has greatly enhanced the clarity and quality of our paper. In response to the reviewer's concerns, we have carefully addressed them as outlined below.
>
> ---
>
> > **R1:
> > While using VR to collect human trajectory data is helpful, this paper would benefit from a discussion in the related works section about VR and human motion. For instance, referencing works like "QuestEnvSim: Environment-aware Simulated Motion Tracking from Sparse Data" in SIGGRAPH 2023 which uses VR for motion tracking and "Strategy and Skill Learning for Physics-based Table Tennis Animation" in SIGGRAPH 2024 which involves interaction between human and humanoid agents.**
>
> **A1:**
> Thank you for your insightful comment. We agree with the reviewer that incorporating the domain of VR and human motion into the related work provides a more comprehensive context for our contributions.
> We have created a new section and included relevant works that analyze human behavior using VR including the two papers suggested by the reviewer after careful consideration. Please see Section.2.3 in our revised manuscript (highlighted in blue).
>
> ---
>
> > **R2:
> I notice authors utilize motion capture to provide whole body motion, and I wonder the reason to consider only experiments of path, trajectory and goal prediction. The occasionally unnatural motion observed in the video could be explained.**
>
> **A2:**
> **Full-body pose is auxiliary information but not our main focus:**
> We incorporated the motion capture system primarily to visualize avatar motions, allowing participants to recognize the movements of others. Consequently, including full-body motion data in the dataset is not our primary focus. In the main paper, we mentioned full-body motion as auxiliary information included in the dataset, noting that head pose (yaw direction) from the raw motion tracking data was incorporated to maximize performance in our evaluation tasks. The head pose offers valuable insights for inferring human intentions, which facilitates more accurate future predictions, as demonstrated in the ablation study (Appendix C).
>
> Additionally, after careful consideration, we have revised the manuscript by replacing 'full-body motion' with alternative terms or explicitly clarifying it as auxiliary data. We believe these revisions better emphasize our primary contribution. The updated sentences in the manuscript are as follows:
>
> - Abstract (L20): Revised "full-body motion" to "accurate trajectory data".
>
> - Introduction (L43): Revised "full-body motions" to "trajectories".
>
> - Figure 1 caption (L89): Revised "full-body motion" to "trajectory".
>
> - Section 3.1 Overview (L154): Revised "it includes full-body human poses, along with head orientation data in addition to trajectories" to "it includes body tracker data on head/waist/hands/feet as auxiliary information".
>
> - Conclusion (L531): Revised "full-body motions" to "accurate trajectory".
>
> All revisions have been highlighted in blue in the manuscript.
>
> **Inaccuracies in avatar motions:**
> As the reviewer noted, avatar motions may occasionally display unnatural joint movements. This issue arises from the performance of the inverse kinematics (IK) software (FINAL-IK), which reconstructs avatar motion using sparse motion trackers placed on the body (head, waist, hands, and feet).
> However, these slight inaccuracies in body motion do not impact the contribution of our experiment, as our focus is on room-scale human dynamics rather than fine-grained body movements.
>
> Our dataset currently includes raw data from sparse motion trackers, which is highly accurate (within a few millimeters). For users requiring precise avatar motion, applying state-of-the-art IK algorithms to the raw tracker data would reconstruct more accurate avatar movements than those displayed in our video.

---

> ### Author Response · Authors · 2024-11-22
> **Response to Reviewer 6iZX (2/3)**
>
> ---
>
> > **R3:
> The use of A\* baselines seems inappropriate for two-person interaction scenarios. I notice this dataset mainly focuses on obstacle avoidance. There appears to be a lack of interactive behaviors between the two persons. It may not be enough if two persons just operate independently and avoid the other person within the same space. I think it doesn't reflect scenarios often seen in real life. Can the authors provide more information on the distribution of action types within the dataset? Given this is a dataset paper, more statistics and descriptions would be beneficial.**
>
> **A3:**
>
> **- On the Use of A\* Baselines**
> The original A* algorithm is not designed to handle two-person interaction scenarios; however, we employed it as a foundational baseline to benchmark basic trajectory prediction and obstacle avoidance capabilities. In contrast, we used A*+Unet baselines to account for social motion behaviors, enabling dataset comparisons. Specifically, A* serves as a deterministic trajectory generator, guided by the probabilistic trajectory distributions produced by Unet models trained on the benchmark datasets. For this reason, we consider A* as a valuable baseline algorithm for our evaluation.
>
> **- Perceived Focus on Obstacle Avoidance**
> While obstacle avoidance is one component, our dataset is not limited to this focus. The trajectories reflect behaviors that go beyond mere avoidance, capturing the nuanced adjustments people make in response to shared space constraints. These include:
> - Dynamic negotiation of personal space in motion.
> - Proximity-based trajectory adjustments that align with real-world social norms.
>
> It is called **social navition**, which has been a hot research topic in the robotics field. We offered a new dataset that contains **two-person trajectories across diverse indoor scenes**, which could make impact on the community. To clarify our contribution on the social navigation, we have modified related works in the manuscript. Please refer Section 2.1 on the manuscript, highlighted in blue color.
>
> For more information on social motion behavior in home environments, please visit [our anonymized website](https://sites.google.com/view/locovr?usp=sharing) to explore typical examples of social motion behaviors featured in LocoVR (Figure 2) and those generated by our trained models (Figure 3). Furthermore, our manuscript (Figure 6 in Appendix E) compares the performance of socially-aware trajectory prediction using single-person and two-person data. The results show that models trained on two-person data successfully predict socially-aware trajectories, while those trained on single-person data do not.
>
> **- Distribution of Action Types and Additional Statistics**
> We appreciate the suggestion to provide a more detailed breakdown of the behaviors captured in the dataset. In the revision, we have included additional statistics, following the prior work on the social navigation dataset [1], which serves as one of the benchmarks in our evaluation.
> We have included **Path efficiency (trajectory complexity)**, **Motion speed**, and **Minimal distance between individuals**, as outlined in the referenced paper. Additionally, we have introduced **Relative speed between individuals** and **Number of speed changes along the trajectory**, to further quantify the characteristics of our dataset.
> The updates are included in the Appendix I.1 on the revised manuscript, highlighted in blue color.
>
> [1]Schreiter, Tim, et al. "THÖR-MAGNI: A large-scale indoor motion capture recording of human movement and robot interaction." The International Journal of Robotics Research (2024).
>
> ---
>
> > **R4:
> The paper seems to focus on locomotion. Without interactions like sitting on a sofa or standing up from a chair, does the goal prediction remain compelling?**
>
> **A4:**
> We consider goal prediction to be one of the crucial tasks in home environments, particularly in its application to human action prediction. For instance, if a person is walking with having a glass, several possible subsequent actions could be inferred based on the scene context: pouring milk at the fridge, placing the glass on the dining table to set up breakfast, bringing it to the couch to hand it to someone, and so on. Our trajectory-based goal prediction approach helps narrow down these candidate actions by predicting target object based on the past trajectory, thereby improving the accuracy of action predictions.
>
> It is important to note that LocoVR is designed to facilitate research on indoor human trajectories and serves as a foundational resource for exploring relationships between motion patterns, goal positions, and indoor scene geometries. While our paper highlights LocoVR's contribution in social navigation, capability of LocoVR is is not limited in that task. We believe LocoVR holds potential for extended applications, such as inferring indoor layouts from trajectory data or studying space utilization in shared environments.

---

> ### Author Response · Authors · 2024-11-22
> **Response to Reviewer 6iZX (3/3)**
>
> ---
>
> > **R5:
> With many figures presented in 2D planes, would a bird's eye view semantic map provide enough information for the prediction tasks? What's the importance of 3D geometry?**
>
> **A5:**
> While our main claim is not on the geometry with 3D and semantic information, we expect these features to enhance the utility of our dataset. To explore this, we conducted a small experiment to evaluate how replacing binary obstacle maps with 3D height maps and semantic maps affects performance.
>
> Table.1 presents the results of the global path prediction task using the UNet+A* model. Each model was trained and tested on LocoVR with binary maps, height maps, and semantic maps, over three trials. The results indicate that models trained with height and semantic maps clearly outperformed those trained with binary maps.
>
> Although we do not yet have a detailed analysis of these findings, they potentially suggest that human trajectories could be influenced by object attributes inferred from height and semantic information. For instance, participants might unconsciously maintain a distance from movable objects, such as chairs or doors, or adjust their trajectories based on the visual clearance provided by different object types. For example, walls, kitchen counters, and low tables offer varying degrees of vision clearance, with lower clearance potentially exerting subtle psychological pressure on trajectory planning.
> A detailed analysis on influence of variational scene information on the human trajectories could provide valuable insights from the perspectives of cognitive and behavioral sciences.
> We have included this result and the discussion in the revised manuscript to mention further potential of our dataset. (Appendix.D.2, highlighted in blue)
>
> Table.1: Accuracy of global path prediction in different range of traveled distance (mean value ± std over 3 trials)
> |              | 0m < d ≤ 3m  | 3m < d ≤ 6m  |    6m < d    |
> |--------------|--------------|--------------|--------------|
> | **binary map**   | 0.138±0.0006 | 0.183±0.0024 | 0.286±0.0113 |
> | **semantic map** | 0.137±0.0004 | 0.170±0.0046 | 0.216±0.0278 |
> | **height map**   | **0.136±0.0011** | **0.165±0.0068** | **0.201±0.0219** |
>
> ---
>
> **[Final Note]** Thank you once again for the insightful review. We believe the revision has strengthened the quality of our paper. If there is anything else we can clarify or elaborate on, please do not hesitate to let us know.

---

> ### Author Response · Authors · 2024-11-25
> **Discussion period approaches its conclusion in the coming days**
>
> **We have done our utmost to address the concerns raised and improve our work based on your valuable comments.**
>
> **If you have any additional questions or points of clarification, we would be delighted to engage in further discussion to ensure that all your concerns are thoroughly resolved.**
>
> **Thank you once again for your invaluable contributions, and we look forward to hearing from you.**

---

> ### Comment · Reviewer_6iZX · 2024-11-26
> **Thank you for the response.**
>
> After reading the rebuttal and other reviews, I have raised my rate to 6.

---

> > ### Author Response · Authors · 2024-11-27
> >
> > Thank you for your thoughtful review and for recognizing our improvements. We greatly appreciate your feedback and the revised score reflecting your positive evaluation.

---

### Official Review · Reviewer_AaG4 · 2024-11-06

**Soundness:** 2
**Presentation:** 3
**Contribution:** 2
**Rating:** 3
**Confidence:** 4

**Summary:**

This paper introduces a dataset for multi-user indoor navigation collected in a Virtual Reality (VR) environment. The dataset includes 7,071 trajectories with 2.5 million frames across 131 scenes. Baseline methods, such as A* and U-Net, are used to demonstrate and analyze the proposed dataset.

**Strengths:**

+ The dataset is based on real human subject studies, and the use of VR environments facilitates data collection efforts.

+ This open-source dataset could benefit the research community focused on social navigation.

**Weaknesses:**

- Novelty and contributions are key concerns. Although the human subject studies require significant time and the dataset a reasonable sample size, the dataset has not been demonstrated, for example, to train state-of-the-art neural network models. The demonstrated methods are relatively simple. The dataset is also limited to two-person navigation scenarios.

- Given that the objective of the dataset is to estimate user trajectories and goal positions, without addressing the estimation of human body motions, why is VR more advantageous than an overhead camera with a bird-eye view?

- How are real human full-body motions in the physical world synchronized with the virtual environment?

-  Research (e.g., in human-robot interaction) has shown that humans respond differently to virtual agents compared to physical agents. The authors are encouraged to provide a study and analysis on whether this difference exists and, if so, how significant it is.

- The work mentions robotics as a motivation and application scenario; however, related research on social robot navigation is not well reviewed.

**Questions:**

See above.

---

> ### Author Response · Authors · 2024-11-22
> **Response to Reviewer AaG4 (1/2)**
>
> We sincerely thank the reviewer for their valuable feedback, which has greatly enhanced the clarity and quality of our paper. In response to the reviewer's concerns, we have carefully addressed them as outlined below.
>
> ---
>
> > **R1: Novelty and contributions are key concerns. Although the human subject studies require significant time and the dataset a reasonable sample size, the dataset has not been demonstrated, for example, to train state-of-the-art neural network models. The demonstrated methods are relatively simple.**
>
> **A1:**
>
> **Evaluation with state-of-the-art models:**
> Although many studies proposed methods to learn human trajectories, most are designed to capture the dynamics of multi-person trajectories in open-space environments and struggle to handle the complexities of indoor navigation due to the limited capability of considering complex room layouts. we employed Ynet [2] as a state-of-the-art benchmark since it is the most recent method capable of predicting human trajectories while accounting for complex indoor geometries, based on the best of our knowledge. Ynet has also been used in recent robotics research [3] as a state-of-the-art benchmark for evaluating trajectory prediction models in indoor scenes. Additionally, we implemented UNet-based models tailored to our specific tasks to compare the performance of the models trained on relevant datasets.
>
> [2]K. Mangalam, et al., "From goals, waypoints and paths to long term human trajectory forecasting." Proceedings of ICCV. 2021.
>
> [3]G. Nicolas, et al., “Long-Term Human Trajectory Prediction Using 3D Dynamic Scene Graphs”, IEEE RA Letters, 2024, 9(12), pp.10978-10985
>
> **Novelty and contribution of our work:**
> Our contribution is the creation of the first dataset that records a wide variety of human trajectories reflecting social motion dynamics within diverse indoor environments. To achieve this diversity efficiently, we utilized a VR-based data collection system. LocoVR is designed to facilitate research on indoor human trajectories and serves as a foundational resource for exploring relationships between motion patterns, goal positions, and indoor scene geometries. Our experiments highlight LocoVR's utility in key tasks, such as socially aware navigation and goal prediction. Additionally, the dataset holds potential for extended applications, such as inferring indoor layouts from trajectory data or studying space utilization in shared environments. By focusing on indoor social motion behaviors, LocoVR provides a unique resource for advancing research on human-centered motion modeling, particularly in confined, interaction-driven settings.
>
> ---
>
> > **R2:
> The dataset is also limited to two-person navigation scenarios.**
>
> **A2:**
>   - **Our goal:** In contrast to conventional studies emphasizing crowd dynamics in open public spaces, our research primarily focuses on social motion behaviors in room-scale private settings. In such environments, individuals exhibit more individualized social behaviors constrained by narrow geometries, such as taking longer detours to avoid others, yielding paths, or maintaining social distance while passing. Given the lack of existing datasets addressing this specific problem setting, our goal is to provide a fundamental resource that is both targeted and extensible, serving as a stepping stone for future datasets that could scale up to include more individuals or different interaction settings.
>
>   - **Two-person navigation scenarios in real-world:** In home environments, interactions involving more than two people are relatively uncommon. Recent census data indicates that 60% of households in the U.S. consist of two people or fewer. Even in households with three or more members, scenarios where more than two individuals navigate a space simultaneously are rare, as residents generally move independently rather than engaging in collaborative or coordinated movements within the private setting. Similarly, other private spaces such as small offices, clinics, or hotel rooms are common examples of everyday environments where two-person navigation is prevalent. This relevance underscores the utility of our dataset for studying interaction dynamics that are directly applicable to these contexts.
>
>   - **Potential utility of two-person navigation dataset:** Two-person interactions are foundational to understanding more complex multi-person dynamics, as they allow us to study detailed interpersonal behaviors such as proxemics, trajectory negotiation, and mutual space adaptation without the confounding variables introduced by larger groups. In the future, researchers can build upon our dataset to study two-person interactions in isolation or as a basis for modeling interactions in more complex, multi-person environments.
>
> We thank the reviewer for highlighting this point, and we included the motivation of two-person setting in the paper.
> (Appendix.I.3, highlighted in blue)

---

> ### Author Response · Authors · 2024-11-22
> **Response to Reviewer AaG4 (2/2)**
>
> ---
>
> > **R3:
> Given that the objective of the dataset is to estimate user trajectories and goal positions, without addressing the estimation of human body motions, why is VR more advantageous than an overhead camera with a bird-eye view?**
>
> **A3:**
> Collecting data in physical home environments is inherently time-intensive, requiring experimenters and participants to travel to the designated location, capture the room layout, set up cameras, and conduct the experiment. Additionally, overhead cameras mounted on ceilings with limited heights are prone to blind spots caused by obstacles, making it challenging to accurately track participants' positions. These challenges have contributed to the lack of diverse indoor locomotion datasets across various scenes.
>
> In contrast, our VR system enables seamless scene switching with a single button click, eliminating the need for physical layout measurements and ensuring precise capture of participants' positions. Thus, collecting two-person trajectory data in VR offers significant advantages, in terms of efficiency and diversity, allowing for the collection of trajectory data across various scenarios in a controlled and repeatable manner.
>
> ---
>
> > **R4:
> How are real human full-body motions in the physical world synchronized with the virtual environment?**
>
> **A4:**
> The 6-point positions/poses of the body (head/waist/hands/feet) are tracked using the HTC VIVE motion capture system. These tracked points are then translated into avatar motions in the VR space through IK (Inverse Kinematics) software (FINAL-IK).
>
> While the avatar motion is primarily used for visualization purposes in our study, we have included the raw motion tracking data (6 points on the body) in the dataset. This data is highly accurate, with deviations within a few millimeters, and is available for further users to reconstruct the avatar motions through state of the art IK algorithms.
>
> ---
>
> > **R5:
> Research (e.g., in human-robot interaction) has shown that humans respond differently to virtual agents compared to physical agents. The authors are encouraged to provide a study and analysis on whether this difference exists and, if so, how significant it is.**
>
> **A5:**
> We think that the impact of the gap between VR and real-world environments varies depending on the task type. For locomotion tasks, we argue that this impact is minimal and does not affect the overall contribution of our dataset for the following reasons:
>
> (1) In our experiment, participants were aware that the virtual avatars were synchronized with real humans sharing the same physical space. This awareness discouraged socially or physically inappropriate behavior, mitigating the potential impact of the VR/real gap, as shown in the studies on VR locomotion[1][2].
>
> (2) Our evaluation used locomotion data collected in physical spaces as test data. Models trained on the LocoVR dataset outperformed those trained on other physically collected datasets (GIMO/THOR-MAGNI), demonstrating that VR-collected data is effective when applied to real-world scenarios. (Tested on LocoReal: Main paper Section.4, tested on GIMO: Appendix D.1)
>
> We have included above discussion in the revised manuscript to clarify the influence of VR/Real gap issue. (Appendix.J, highlighted in blue)
>
> [1] H. Yun, Y. Watanabe, A. Yamada, "Exploring the Role of Expected Collision Feedback in Crowded Virtual Environments," Proc. IEEE Conf. Virtual Reality and 3D User Interfaces, 2024.
>
> [2] A. L. Simeone, I. Mavridou, and W. Powell, "Altering user movement behaviour in virtual environments," IEEE Transactions on Visualization and Computer Graphics, vol. 23, no. 4, pp. 1312–1321, 2017.
>
> ---
>
> > **R6:
> The work mentions robotics as a motivation and application scenario; however, related research on social robot navigation is not well reviewed.**
>
> **A6:**
> We appreciate the reviewer for this insightful suggestion. We agree with the reviewer that incorporating the domain of social robot navigation into the related work provides a more comprehensive context for our contributions. We have added works relevant to social robot navigation in the related works section. (Section 2.1, highlighted in blue)
>
> ---
>
> **[Final Note]** Thank you once again for the insightful review. We believe the revision definitely has strengthened the quality of our paper. If there is anything else we can clarify or elaborate on, please do not hesitate to let us know.

---

> ### Author Response · Authors · 2024-11-25
> **Discussion period approaches its conclusion in the coming days**
>
> **We have done our utmost to address the concerns raised and improve our work based on your valuable comments.**
>
> **If you have any additional questions or points of clarification, we would be delighted to engage in further discussion to ensure that all your concerns are thoroughly resolved.**
>
> **Thank you once again for your invaluable contributions, and we look forward to hearing from you.**

---

> ### Author Response · Authors · 2024-11-27
> **24 hours remaining before the paper revision deadline**
>
> Dear Reviewer AaG4,
>
> We hope this message finds you well. With 24 hours remaining before the revision deadline, we kindly request your feedback on the remaining concerns based on our responses. We understand the review process is time-consuming, but your feedback is invaluable in shaping the final outcome. Thank you again for your time and effort in reviewing our work.

---

### Author Response · Authors · 2024-11-23
**Common response to Reviewers and the Area Chair (3/3)**

---

## **Additional experiment results:**

---

- **Influence of scene information types - Difference in performance with Binary obstacle map / Semantic map / Height map:**

While our main claim is not on the geometry with 3D and semantic information, we expect these features to enhance the utility of our dataset. To explore this, we conducted a small experiment to evaluate how replacing binary obstacle maps with 3D height maps and semantic maps affects performance.

Table.1 presents the results of the global path prediction task using the UNet+A* model. Each model was trained and tested on LocoVR with binary maps, height maps, and semantic maps, over three trials. **The results indicate that models trained with height and semantic maps clearly outperformed those trained with binary maps.**

Although we do not yet have a detailed analysis of these findings, they **potentially suggest that human trajectories could be influenced by object attributes inferred from height and semantic information**. For instance, participants might unconsciously maintain a distance from movable objects, such as chairs or doors, or adjust their trajectories based on the visual clearance provided by different object types. For example, walls, kitchen counters, and low tables offer varying degrees of vision clearance, with lower clearance potentially exerting subtle psychological pressure on trajectory planning. A detailed analysis on influence of variational scene information on the human trajectories could provide valuable insights from the perspectives of cognitive and behavioral sciences. We have included this result and the discussion in the revised manuscript to mention further potential of our dataset. (Appendix.D.2, highlighted in blue)

Table.1: Accuracy of global path prediction in different range of traveled distance (mean value ± std over 3 trials)
|              | 0m < d ≤ 3m  | 3m < d ≤ 6m  |    6m < d    |
|--------------|--------------|--------------|--------------|
| **binary map**   | 0.138±0.0006 | 0.183±0.0024 | 0.286±0.0113 |
| **semantic map** | 0.137±0.0004 | 0.170±0.0046 | 0.216±0.0278 |
| **height map**   | **0.136±0.0011** | **0.165±0.0068** | **0.201±0.0219** |

---

- **Demographics in participants and variations in locomotion pairs:**

There were 32 participants in total, comprising 21 males and 11 females, with ages ranging from 18 to 42. From this pool, pairs were formed to conduct 25 experiments, each involving a unique pair (Table.2). The experiments included various combinations of male-male, female-female, and male-female pairs, as well as pairs of friends and nonfriends, as shown in the table.3.

As the reactions between pairs in close proximity are influenced by attributes and interpersonal relationships, **further data analysis may provide new insights into the relationship between these attributes, relationships, and behavioral patterns**. It could be an intriguing study from the perspective of cognitive and behavioral sciences. We have added the information shown in this reply in the revised manuscript. (Appendix.I.2, highlighted in blue)

Table.2: User demographics (categorized by gender and age)
|     Age    |   Male   |  Female  |
|------------|----------|----------|
|  **under 20**  |     5    |     3    |
|  **20 to 29**  |     15   |     8    |
|   **over 30**  |     1    |     0    |


Table.3: Diversity of pairs (categorized by gender and relationship)
|              |  Male-Male  | Female-Female | Male-Female |
|--------------|-------------|---------------|-------------|
|    **Friends**   |       2     |       2       |      5      |
|  **Non-friends** |       9     |       1       |      6      |


---

- **Additional data statistics in LocoVR:**

In the revision, we have included additional statistics, following the prior work on the social navigation dataset [1], which serves as one of the benchmarks in our evaluation. We have included **Path efficiency (trajectory complexity)**, **Motion speed**, and **Minimal distance between individuals**, as outlined in the referenced paper. Additionally, we have introduced **Relative speed between individuals** and **Number of speed changes along the trajectory**, to further quantify the characteristics of our dataset. The updates are included in the Appendix I.1 on the revised manuscript, highlighted in blue color.

[1]Schreiter, Tim, et al. "THÖR-MAGNI: A large-scale indoor motion capture recording of human movement and robot interaction." The International Journal of Robotics Research (2024).

---

### Author Response · Authors · 2024-11-23
**Common response to Reviewer and the Area Chair (2/3)**

---

- **What is the motivation of the problem setting? Why two-person?:**
We agree that variations in locomotion patterns involving more than two individuals may occur in real-world scenarios, particularly in open public spaces. However, our research focuses on private indoor settings where the number of pedestrians is typically very limited, and individuals generally move independently. We believe this focus does not diminish the contribution of our dataset, as scenarios in private indoor settings are both common and essential in real-world contexts, yet they have been largely overlooked by most existing datasets. Please see the following for a detailed discussion.

  - **Our goal:** In contrast to conventional studies emphasizing crowd dynamics in open public spaces, our research primarily focuses on social motion behaviors in room-scale private settings. In such environments, individuals exhibit more individualized social behaviors constrained by narrow geometries, such as taking longer detours to avoid others, yielding paths, or maintaining social distance while passing. Given the lack of existing datasets addressing this specific problem setting, our goal is to provide a fundamental resource that is both targeted and extensible, serving as a stepping stone for future datasets that could scale up to include more individuals or different interaction settings.

  - **Two-person navigation scenarios in real-world relevance:** In home environments, interactions involving more than two people are relatively uncommon. Recent census data indicates that 60% of households in the U.S. consist of two people or fewer. Even in households with three or more members, scenarios where more than two individuals navigate a space simultaneously are rare, as residents generally move independently rather than engaging in collaborative or coordinated movements within the private setting. Similarly, other private spaces such as small offices, clinics, or hotel rooms are common examples of everyday environments where two-person navigation is prevalent. This relevance underscores the utility of our dataset for studying interaction dynamics that are directly applicable to these contexts.

  - **Potential utility of two-person navigation dataset:** Two-person interactions are foundational to understanding more complex multi-person dynamics, as they allow us to study detailed interpersonal behaviors such as proxemics, trajectory negotiation, and mutual space adaptation without the confounding variables introduced by larger groups. In the future, researchers can build upon our dataset to study two-person interactions in isolation or as a basis for modeling interactions in more complex, multi-person environments.

---

- **What is the role of full-body motion in our work?:**

We incorporated the motion capture system primarily to visualize avatar motions, allowing participants to recognize the movements of others. Consequently, **including full-body motion data in the dataset is not our primary focus**. In the main paper, we mentioned full-body motion as **auxiliary information included in the dataset**, noting that head pose (yaw direction) from the raw motion tracking data was incorporated to maximize performance in our evaluation tasks. The head pose offers valuable insights for inferring human intentions, which facilitates more accurate future predictions, as demonstrated in the ablation study (Appendix C).

---

- **Why inaccuracy in avatar motion occur?:**

As seen in video, avatar motions may occasionally display unnatural joint movements. **This issue arises from the performance of the inverse kinematics (IK) software (FINAL-IK)**, which reconstructs avatar motion using sparse motion trackers placed on the body (head, waist, hands, and feet). However, these slight inaccuracies in body motion **do not impact the contribution of our experiment**, as our focus is on room-scale human dynamics rather than fine-grained body movements.

Our dataset currently includes raw data from sparse motion trackers, which is highly accurate (within a few millimeters). For users requiring precise avatar motion, applying state-of-the-art IK algorithms to the raw tracker data would reconstruct more accurate avatar movements than those displayed in our video.

---

### Author Response · Authors · 2024-11-23
**Common response to Reviewer and the Area Chair (1/3)**

## **Dear Reviewers and the Area Chair,**

We sincerely appreciate the time and effort you dedicated to reviewing our work. We have made our best effort to address the reviewers' concerns and improve the quality of the paper. To facilitate a clearer understanding of our work, we provide a shared Q&A section for the reviewers and the area chair, along with additional experimental results.

---

## **General questions:**

---

- **What is the novelty and contribution of our work?:**

Our contribution is the creation of the first dataset that records **a wide variety of human trajectories reflecting social motion dynamics within diverse indoor environments**. To achieve this diversity efficiently, we utilized a VR-based data collection system. LocoVR is designed to facilitate research on indoor human trajectories and **serves as a foundational resource for exploring relationships between motion patterns, goal positions, and indoor scene geometries**. Our experiments highlight LocoVR's utility in key tasks, such as socially aware navigation and goal prediction. Additionally, the dataset holds potential for extended applications, such as inferring indoor layouts from trajectory data or studying space utilization in shared environments. By focusing on indoor social motion behaviors, LocoVR provides a unique resource for advancing research on human-centered motion modeling, particularly in confined, interaction-driven settings.

---

- **What is the advantage in collecting locomotion data in VR?:**

**Challenge in indoor scenes:**
Collecting data in physical home environments is **inherently time-intensive, requiring experimenters and participants to travel to the designated location, capture the room layout, set up cameras**, and conduct the experiment. Additionally, overhead cameras mounted on ceilings with limited heights are prone to blind spots caused by obstacles, making it **challenging to accurately track participants' positions**. **These challenges have contributed to the lack of diverse indoor locomotion datasets across various scenes.**

**Advantage in VR:**
In contrast, our VR system enables **seamless scene switching with a single button click, eliminating the need for physical layout measurements and ensuring precise capture of participants' positions**. Thus, collecting two-person trajectory data in VR offers significant advantages, in terms of **efficiency and diversity**, allowing for the collection of trajectory data across various scenarios in a controlled and repeatable manner.

---


- **Is there any influence of the gap between VR/Real on the dataset?:**

We think that the impact of the gap between VR and real-world environments varies depending on the task type. For locomotion tasks, we argue that this impact is minimal and does not affect the overall contribution of our dataset for the following reasons:

(1) In our experiment, **participants were aware that the virtual avatars were synchronized with real humans sharing the same physical space**. Also, **the avatar enables participants to percept relative position between their body and surrounding objects**. These awareness **discouraged socially or physically inappropriate behavior, mitigating the potential impact of the VR/real gap**, as demonstrated in a recent study on VR locomotion[1][2]. In addition, we have introduced a filter to detect instances of users passing through virtual objects to remove such data from the dataset.

(2) Our evaluation used locomotion data collected in physical spaces as test data. Models trained on the LocoVR dataset outperformed those trained on other physically collected datasets (GIMO/THOR-MAGNI), demonstrating that **VR-collected data is effective when applied to real-world scenarios.**

[1] H. Yun, Y. Watanabe, A. Yamada, "Exploring the Role of Expected Collision Feedback in Crowded Virtual Environments," Proc. IEEE Conf. Virtual Reality and 3D User Interfaces, 2024.

[2] A. L. Simeone, I. Mavridou, and W. Powell, "Altering user movement behaviour in virtual environments," IEEE Transactions on Visualization and Computer Graphics, vol. 23, no. 4, pp. 1312–1321, 2017.

---

### Author Response · Authors · 2024-12-03
**Summary of the review (2/2)**

# **Summary of the review: Common concerns raised by the reviewers**

**1. The gap between VR and real-world settings could influence the motion behavior of participants (Reviewer AaG4, Reviewer 1K21, Reviewer PhCB):**

- **Short answer:** We think that the impact of the gap between VR and real-world environments varies depending on the task type. For locomotion tasks, we argue that this impact is minimal and does not affect the overall contribution of our dataset for the following reasons: (1) the use of a real human-based experimental setting, and (2) the demonstrated performance on real-world data.
**Detailed discussions are described in the response to each reviewer.**

**2. Focus on two-person navigation may limit its applicability in real-world scenarios. (Reviewer AaG4, Reviewer Fpz9, Reviewer PhCB):**

- **Short answer:** We agree that variations in locomotion patterns involving more than two individuals may occur in real-world scenarios, particularly in open public spaces. However, our research focuses on private indoor settings where the number of pedestrians is typically very limited, and individuals generally move independently. We believe this focus does not diminish the contribution of our dataset, as scenarios in private indoor settings are both common and essential in real-world contexts, yet they have been largely overlooked by most existing datasets. **Detailed discussions are described in the response to each reviewer.**

**3. Role of the full-body motion is not clear (Reviewer 6iZX, Reviewer PhCB):**

- **Short answer:** We incorporated the motion capture system primarily to visualize avatar motions, allowing participants to recognize the movements of others. Consequently, full-body motion data is auxiliary information and not our primary focus of our dataset. **Detailed discussions are described in the response to each reviewer.**

**4. Avatar motions in the video partly look unnatural (Reviewer 6iZX, Reviewer PhCB):**

- **Short answer:** Unnatural joint movements in avatar motions occasionally appear due to the limitations of the inverse kinematics (IK) software (FINAL-IK) performance, however, these minor inaccuracies do not impact the contribution of our experiment, as our focus is on room-scale human dynamics rather than fine-grained body movements. **Detailed discussions are described in the response to each reviewer.**

---

## **Paper revisions:**

---

- **Newly included discussions:**
  - Discussion on the influence of the gap between VR/Real: Appendix J  (Reviewer AaG4, Reviewer 1K21, Reviewer PhCB)
  - Discussion on the motivation of our problem setting (two-person locomotion): Appendix I.3  (Reviewer AaG4, Reviewer Fpz9, Reviewer PhCB)
  - Discussion on the occasional inaccuracy in avatar motion: Appendix F.5 (Reviewer 6iZX, Reviewer PhCB)
  - Description on the prior works on the "robot social navigation" in the related works: Section 2.1 (Reviewer AaG4)
  - Description on the prior works on the "VR-based human behavior analysis" in the related works: Section 2.3 (Reviewer 6iZX)
  - Discussion on potential future applications in the future work: Section 5 (Reviewer Fpz9)
  - Description on the motivation of parameter settings in data augmentation: Appendix H (Reviewer PhCB)

- **Newly included data:**
  - Influence of scene information types - Difference in performance with Binary obstacle map / Semantic map / Height map: Appendix D.2 (Reviewer 6iZX)
  - Demographics in participants and variations in locomotion pairs: Appendix I.2 (Reviewer 1K21)
  - Additional data statistics in LocoVR: Appendix I.1 (Reviewer 6iZX)

- **Modifications:**
  - Modified the statements on the "full-body motion" to clarify it is auxiliary information in the dataset, not our prior focus: (Reviewer 6iZX, Reviewer PhCB)
    - Abstract (L20): Revised "full-body motion" to "accurate trajectory data".
    - Introduction (L43): Revised "full-body motions" to "trajectories".
    - Figure 1 caption (L89): Revised "full-body motion" to "trajectory".
    - Section 3.1 Overview (L154): Revised "it includes full-body human poses, along with head orientation data in addition to trajectories" to "it includes body tracker data on head/waist/hands/feet as auxiliary information".
    - Conclusion (L531): Revised "full-body motions" to "accurate trajectory".
  - Modified of a typo: Appendix D (Reviewer PhCB)

---

---

### Author Response · Authors · 2024-12-03
**Summary of the review (1/2)**

# **Summary of the review: Key strengths highlighted by the reviewers**

**1. Significance of the dataset:**
- Introduction of motion proxemics in a large-scale dataset which is **useful for downstream tasks** such as studying human-human interactions and potentially human-robot interactions. (Reviewer 1K21)
- Data of two-person trajectories is **useful to study motion proxemics** (Reviewer 1K21)
- The extensive collection of two-person human trajectory (with motion capture) data in indoor scenes, is a **valuable resource for the community**. (Reviewer 6iZX)
- This open-source dataset could **benefit the research community focused on social navigation**. (Reviewer AaG4)
- I can see implications of this work not just for virtual agents, games etc, but also **as we move to further robotic presence in our homes this type of dataset can help train their trajectories**. (Reviewer Fpz9)
- This paper captures two-person goal-reaching motions which include the social navigation behaviors such as adjusting the path to respect personal boundaries or side steps to give way to another person. Such social navigation behavior is **not covered in most previous datasets** and is **important for understanding multi-human social navigation and potential human-robot interactions.** (Reviewer PhCB)

**2. Significance of the VR-based data collection approach:**
- Utilizing virtual reality for data collection is a **promising approach**, allowing more diverse 3D scenes when resources are limited. (Reviewer 6iZX)
- The dataset is based on real human subject studies, and the use of VR environments **facilitates data collection efforts**. (Reviewer AaG4)
- The proposed VR capture solution **eliminates the high cost of physically setting up indoor scenes** and capturing human movements, which facilitates scaling up locomotion capture to many more scenes compared to previous datasets. (Reviewer PhCB)

**3. Validity of the evaluation:**
- **Rigorously quantitatively evaluated against strong baselines**; **baseline configurations and settings are fair and well-documented** (Reviewer 1K21)
- **The dataset is well evaluated.** The paper evaluates the dataset on three trajectory-based tasks—global path prediction, trajectory prediction, and goal prediction. The results show that models trained on LocoVR outperform those trained on other datasets, particularly in predicting realistic, socially aware navigation paths in complex environments. (Reviewer Fpz9)

**4. Other comments:**
- The authors introduce an **exciting/ relevant problem**. The problem is **well-motivated while the execution and evaluations are strong**. I see **many potential downstream applications** and I believe that this will make a **huge impact in the robotics field**. (Reviewer 1K21)

---

### Meta-Review · Area_Chair_xEea · 2024-12-20

**Metareview:**

The submission is about a new dataset of two-person trajectories captured in VR.  Reviewers acknowledged the usefulness of the dataset; they also raised some concerns, primarily about the validity of data collected in VR.  Post rebuttal, most reviewers were convinced and supported acceptance.  Reviewer AaG4 remained negative but did not engage in discussions.  The AC agreed with the majority and recommended acceptance.  The authors should revise the submission in the camera ready to address the remaining concerns.

**Additional Comments On Reviewer Discussion:**

The discussion convinced most reviewers to support the submission's acceptance.

---

### Decision · Program_Chairs · 2025-01-22

Accept (Poster)